

# Reconstruction of Arctic sea ice thickness and its impact on sea ice forecasting in the melting season

Lu Yang[1], Hongli Fu[2], Xiaofan Luo[1], Shaoqing Zhang[3], Xuefeng Zhang[1]

[1]School of Marine Science and Technology, Tianjin University, Tianjin, 300072, China
[2]Key Laboratory of Marine Environmental Information Technology, National Marine Data and Information Service, Ministry of Natural Resources, Tianjin, 300171, China
[3]Key Laboratory of Physical Oceanography, Ministry of Education, Ocean University of China, Qingdao, 266100, China

*Correspondence to*: Xuefeng Zhang (xuefeng.zhang@tju.edu.cn)

**Abstract.** Generally, the sea ice prediction skills can be improved via assimilating available observations of the sea ice
concentration (SIC) and the sea ice thickness (SIT) into a numerical forecast model to update the initial fields of the model. However, due to the lack of SIT satellite observations in the melting season, only SIC fields in the forecast model can be directly updated, which will bring about the dynamical mismatch between SIC and SIT to affect the model prediction accuracy. In order to solve this problem, a statistically based bivariate regression model of SIT, named as BRMT, is tentatively established based on the grid reanalysis data of SIC and SIT, to reconstruct the daily Arctic sea ice thickness data.
Both BRMT-constructed SIT and several popular reanalysis datasets are compared to each other and validated based on available SIT observations in situ. Results show that BRMT can effectively reproduce the spatial and temporal changes of ice thickness in the melting season, and BRMT-constructed SIT is more accurate in capturing the change trend of ice thickness over a period of time, also the reconstructed SIT of one-year ice and multi-year ice types in the central Arctic and E Greenland Sea are closer to the observations. Further, as SIT from BRMT and SIC from satellite remote sensing are jointly
assimilated into the ice-sea coupled numerical model, the prediction accuracy of SIC and SIT in the Arctic melting season is significantly improved, especially the SIC in the marginal ice zone and SIT in the central Arctic.

## 1 Introduction

The Arctic is one of the most important regions for the exchange of materials and energy between the atmosphere and the ocean, and the major interaction between the Arctic Ocean and the global climate system is reflected through the sea ice.
However, observations in the past 30 years have shown that Arctic sea ice is undergoing rapid changes (Kwok and Cunningham, 2015). From 1979 to 2017, the sea ice extent decreased by 3.24 million square kilometres in September, with a significant decrease in the Arctic sea ice margin from the Beaufort Sea in the west to the Barents Sea in the north (Liu et al., 2019). As the ice shrinks, the ice floes are thinning. Recent satellite data show an average reduction of about 50% in Arctic sea ice thickness (SIT) compared to submarine sea ice observations during 1958-1976 (Kwok and Rothrock, 2009). The
above changes in Arctic sea ice have aroused people's attention and posed major opportunity to Arctic maritime activities,



such as polar shipping, fishing and oil/gas resources exploration. Meanwhile, the grasp of sea ice concentration (SIC), SIT and other information is crucial for the polar research (Takuya et al., 2018). Therefore, accurate real-time sea ice prediction has become an urgent need (Eicken, 2013). It is well known that Arctic sea ice numerical models used for the synoptic-scale forecasting depend heavily on the initial state fields of the model. Therefore, it is necessary to integrate the sea ice

observations into the numerical forecast model using appropriate data assimilation methods to generate a more realistic initial condition and improve the prediction ability of Arctic sea ice (Lisæter et al., 2003).

Lindasy and Zhang (2006) used Nudging method to assimilate SIC observation. The assimilated SIC improves the match with the observed extent, but the sea ice draft in the Fram Strait is underestimated by 0.64 m compared with the available observations. Lisæter et al. (2003) assimilated SIC observation data using the Ensemble Kalman Filter (EnKF) (Evensen,

1994) based on the coupled ice-sea model. Although the correlation coefficient between the forecast SIC and SIT reaches more than 0.5 in winter, it drops below 0.3 or even reaches a negative value in the melting season. Wang et al. (2013) proposed a method combining optimal interpolation with the Nudging to assimilate SIC. Results show that there are significant improvements for SIC analysis result in the sea ice margin region in summer, but there are deviations in the prediction of sea ice extent. Yang et al. (2015a) used Local Singular Evolution Interpolation Kalman Filter (LSEIK) to

assimilate SIC in summer. The consistency of forecast SIC with satellite observations is improved, but the multi-year ice thickness in the central Arctic is overestimated by more than 1 m.

Day et al. (2014) showed that accuracy of initial SIT is also important for the prediction of SIC and sea ice extent in summer. Lisæter et al. (2007) used EnKF to assimilate the SIT detected by Cryosat satellite and improved the quality of initial SIT forecast field. The results showed the prediction accuracy of SIC, sea surface temperature (SST) and sea surface salinity are

well improved by the SIT assimilation. Yang et al. (2014) used LSEIK to simultaneously assimilate Special Sensor Microwave Imager/Sounder (SSMIS) SIC and Soil Moisture and Ocean Salinity (SMOS) SIT in the cold season. Compared with only SIC assimilation or no data assimilation, the root mean square error (RMSE) of SIT forecast results is reduced 0.47 m. However, SMOS SIT observation data are only applicable to thin ice (<1 m) (Tian-Kunze et al., 2014), the assimilation of which only improves the one-year ice prediction in the marginal area of sea ice, while the thick (multi-year) ice cannot be

significantly improved during the early melting and freezing seasons (Yang et al., 2016; Xie et al., 2016). In cold season, Mu et al. (2018b) not only assimilated SMOS SIT and SSMIS SIC observation data, but also simultaneously assimilated Cryosat-2 SIT observation data which can better capture interannual changes of thick ice areas (Laxon et al., 2013). By using the complementary characteristics of the two kinds of thickness data, the overall RMSE of SIT and SIC is smaller than the counterpart under the condition of assimilating both SMOS SIT and SSMIS SIC data.

It is worth noting that there are few studies on the prediction of SIT during the melting season due to the sparse in situ SIT observations and insufficient satellite remote sensing SIT observations (Ricker et al., 2014; Ricker et al., 2017). In order to solve this problem, Mu et al. (2018a) combined the skill of satellite thickness assimilation in the freezing season with the model skill in the melting season, and a combined model and satellite thickness (CMST) is proposed to estimate the thickness of Arctic sea ice in the melting season. Yang et al. (2019) used the restart files from the CMST system as the sea



ice initial condition, which assimilated the SIC from satellite remote sensing and the SIT from SMOS and CryoSat-2. Based on this, an integrated sea ice seasonal prediction system (SISPS) is designed for the real-time prediction of summer sea ice conditions in the Arctic. CMST estimation is heavily dependent on the quality of satellite data products and the parameterization schemes of physical processes in the model, which has certain uncertainties (Mu et al., 2018a). Chi and Kim (2021) proposed an ensemble one-dimensional convolutional neural network using the deep learning (DL) approach. It

concatenates the input features of multiple Advanced Microwave Scanning Radiometer 2 (AMSR2) channel types generated by mathematical operation to estimate SIT with a resolution of 25 km. However, due to the high uncertainties of summer SIT retrievals, some researchers including Chi only estimated the SIT for the freezing season (October to April) (Kaleschke et al., 2012; Lee et al., 2020).

In this study, based on the strong positive correlation between SIC and SIT from the perspective of statistical significance

(Yang et al., 2015b), we proposed a bivariate regression model of SIT (BRMT) to obtain the SIT records in the melting season covering the whole Arctic region. To conform to the realistic sea ice state, we consider the multi-year historical reanalysis data of SIC and SIT in the melting season, and construct the correlation between them. Then, using satellite remote sensing observation data of SIC and the relationship between them, the corresponding "pseudo" observation field of SIT can be reconstructed. Finally, based on the sea-ice coupled model, the spatial multiscale recursive filter (SMRF) data

assimilation method is used to jointly assimilate the SIC observations from satellite remote sensing and the SIT "pseudo" observations from BRMT, so as to generate a more real sea ice initial field and obtain higher prediction accuracy of sea ice variables.

The paper is organized as follows: In Sect. 2, we describe the observation data, reanalysis data and other SIT data sets used to evaluate the SIT regression model. Sect. 3 focuses on the BRMT, briefly introduces the numerical model and assimilation

method used in the prediction experiments, and gives the evaluation criteria. The comparisons among the reconstructed SIT via BRMT, in situ observations and several reanalysis datasets are presented in Sect. 4. On this basis, the real-time Arctic sea ice numerical forecast experiments are carried out in Sect. 5. Finally, the discussion and conclusion of this study are drawn in Sect. 6.

## 2 Data sources

### 2.1 Observation data

In this work, SIC refers to the proportion of sea ice covered area in unit space, the variation range of which is 0-1. The daily SIC observations are derived from the daily passive microwave data of the special sensor microwave/imager (SSMI) carried by DMSP F-17. It was processed by National Snow and Ice Data Centre (NSIDC) with NASA team algorithm (Cavalieri et al., 2012). The spatial resolution is 25 km×25 km.

In addition, to assess the reconstructed SIT and predicted SIT, two type of in situ SIT observations are used. The first is the sea ice draft, which comes from Upward Looking Sonar (ULS) measurements of Beaufort Gyre Experiment Program



(BGEP). It can be converted into thickness by multiplying by 1.1, which is approximately equal to the ratio of mean seawater density of 1024 kg m$^{-3}$ and sea ice density of 910 kg m$^{-3}$ (Nguyen et al., 2011). The error of ULS measurements of ice draft is estimated about 0.1 m (Melling et al., 1995). The other is the SIT data, which is derived from Ice Mass balance

Buoys (IMB) deployed on the surface of Arctic sea ice (Perovich et al., 2022). The two acoustic rangefinders on the IMB monitor the position of the ice bottom and the snow and as well as the ice surface, which is used to estimate the SIT. The accuracy of both detectors is 5 mm (Richter-Menge et al., 2006). In this study, the data are selected from mooring facilities of the BGEP_A located in (74° 59.816′ N, 149° 58.149′ W), BGEP_B located in (78° 0.395′ N, 149° 58.462′ W), BGEP_D located in (73° 59.649′ N, 139° 59.043′ W) and 25 IMB buoys with available SIT during the melting season from 2011 to

105 2015.

### 2.2 Other data sets

The reanalysis data used in the regression model are from TOPAZ4 version of the Nansen Centre for Environment and Remote Sensing, Norway's Sea Ice/Ocean Numerical Prediction System (Xie et al., 2017). The data set is named Arctic_Reanalysis_Phys_002_003. It includes daily SIC, SIT and sea ice velocity. The spatial resolution is 12.5km×12.5km,

the time range is from 1 January 1991 to 31 December 2019, and the region covers the Arctic Ocean.

Besides, three additional SIT data sets are selected as the contrasts to more comprehensively validate the accuracy of the reconstructed SIT. The first is the Arctic SIT record of CMST, which is generated via assimilating weekly averaged Cryosat-2 SIT, daily SMOS SIT and daily Special Sensor Microwave Imager Sounder SIC (Mu et al., 2018a) into the Massachusetts Institute of Technology general circulation model (MITgcm), using a local Error Subspace Transform Kalman filter

(LESTKF) coded in the parallel data assimilation framework (PDAF). The CMST (v1.0) are provided north of 65° N from 1 October 2010 to 31 December 2016. The other two come from Pan-Arctic Ice-Ocean Modeling and Assimilation System (PIOMAS), and Global Ice-Ocean Modeling and Assimilation System (GIOMAS) (Zhang and Rothrock, 2003), both of which are composed of global Parallel Ocean and sea Ice Models (POIM) with data assimilation capability. POIM couples the Parallel Ocean Program and the Thickness and Enthalpy Distribution sea ice model. The atmospheric information used to

drive the PIOMAS and GIOMAS are from NCEP/NCAR reanalysis data, including the wind, the surface air temperature and the cloud cover to compute solar and long wave radiation. In addition, sea ice concentration information from the NSIDC near real time product are assimilated into the model to improve ice thickness estimates. In this study, the daily averaged SIT of PIOMAS V2.1 and the monthly averaged SIT of GIOMAS are selected respectively, and the time range is 1 July to 30 September from 2011 to 2015.





## 3 Method

### 3.1 Bivariate regression model of SIT

Considering that it is impossible to obtain Arctic SIT observation from the satellites remote sensing in the melting season since the most advanced inversion algorithm was impeded by the saturated surface water vapor from the surface snow melt (Ricker et al., 2017), we focus on the reconstruction of SIT in the melting season (boreal July-September). Meanwhile, in order to facilitate the comparison of the reconstructed SIT with those from other data sets and in situ observation, the TOPAZ4 reanalysis from 2011 to 2015 is chosen as an example to develop the bivariate regression model for SIT.

First of all, the TOPAZ4 reanalysis of the SIC and SIT from 2004 to 2018 are selected to calculate the daily spatial correlation coefficients of SIC and SIT in the melting season. According to the scatter plots between correlation coefficients and dates (Fig. 1), except for a few dates, most of the correlation coefficients between SIC and SIT in all years are between 0.80 and 0.95, which completely pass the significance test. It indicates that there is a strong correlation between SIC and SIT, which provides the theoretical support for the establishment of regression model in the next step.

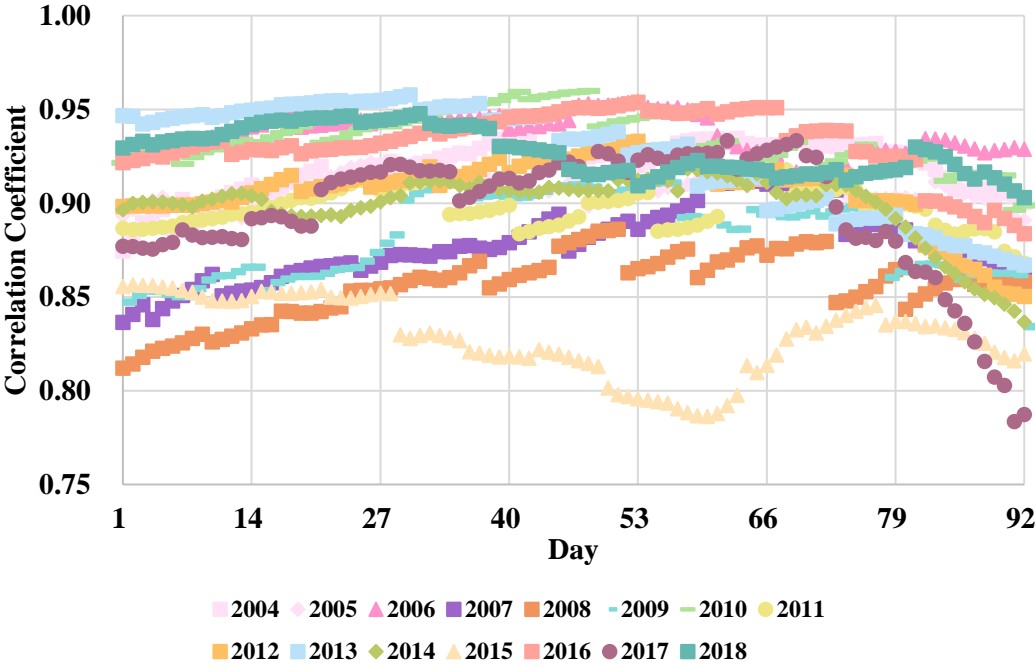

**Figure 1.** Spatial correlation coefficient between daily sea ice concentration and sea ice thickness during melting season from 2004 to 2018.

In order to match the reanalysis data grid (609×881) with the observation data grid (304×448), data pre-processing is required to project the reanalysis data of SIC and SIT into the observation grid in the melting season (92 days in total) from 2004 to 2018. In order to illustrate the construction process of the model, 1 July 2011 is chosen as an example (the flow chart





is shown in Fig. 2). Firstly, the reanalysis data of daily SIC and SIT are selected on 1 July in 14 years from 2004 to 2010 and from 2012 to 2018. In order not to introduce a priori knowledge, the reanalysis data of 2011 is removed from the selection of

years. Then, the linear regression process is carried out at each grid point of the whole region (non null point) for each year, with SIC being as the independent variable and SIT being as the dependent variable. The corresponding SIC-SIT regression relation at each grid point can be obtained for each year.

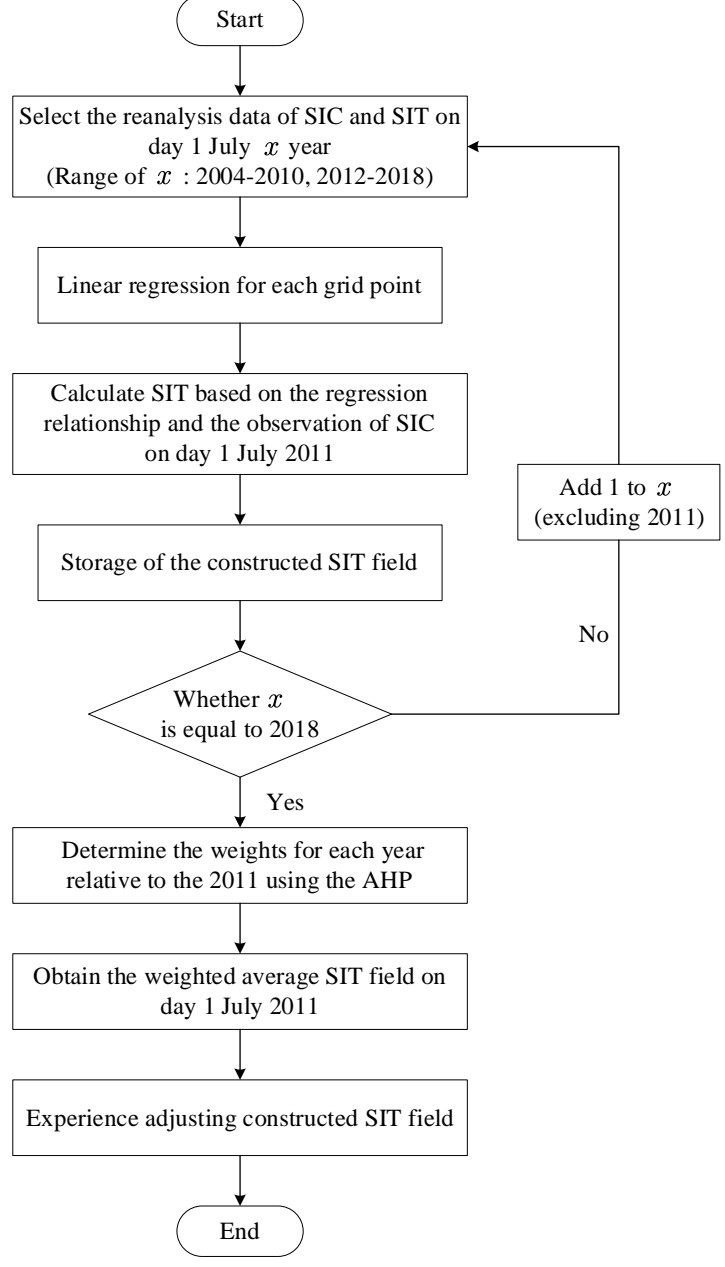

**Figure 2.** A flow chart of bivariate regression model of SIT (take 1 July 2011 as an example).





The SIT at each grid point at each year can be calculated from the fitting relation and the SIC observational data on 1 July 2011, that is, 14 groups of SIT fields are formed on 1 July 2011. Then, the 9-quartile scale method in the Analytic Hierarchy Process (AHP) (Rahman & Frair, 1984) is used to determine the weight of each year relative to 2011, as shown in Table A1 in the appendix (all results pass the consistency test). The constructed SIT field on 1 July 2011 is obtained by the weighting average. The construction process of thickness field on other days in 2011 is the same as the process mentioned above. It is

worth noting that the selection of reanalysis data time period in the process of reconstructing SIT in other years is slightly different from that in 2011. In order to avoid the target year of the current construction process, the years selected for the model calculation in different target years are different. The details can be seen in Table A1 in the appendix.

Finally, according to the SIC observational data, the SIT field is empirically adjusted to make the constructed SIT consistent with the SIC (Preller et al., 2002): if the value of SIC at a certain grid point is 0 and the constructed SIT at this grid point is

not 0, the ice will be removed from the constructed SIT field; If the value of constructed SIT at a certain grid point is 0 and the SIC at this grid point is not 0, the constructed SIT is adjusted to 1.0 m (if SIC > 0.5) or 0.5 m (if SIC < 0.5). The empirically adjusted SIT fields in the melting season from 2011 to 2015 is the reconstructed SIT via BRMT.

### 3.2 Numerical model and data assimilation method

### 3.2.1 The ice-ocean coupled model

The used ocean model in this study is the MITgcm (Marshall et al., 1997), which solves the three-dimensional primitive equations with implicit linear free-surface under the hydrostatic and Boussinesq approximations. The ocean model is coupled to a sea-ice model that computes ice thickness, ice concentration, and snow cover as Zhang et al. (1998) and that simulates a viscous-plastic rheology using an efficient parallel implementation of the Zhang and Hibler (1997) solver. The coupled model used in this study adopts a global cubic spherical grid, and the Arctic region includes 510×510 grid points with an

average horizontal distance of 18 km. The open boundary is about 55° N in the Atlantic Ocean and Pacific Ocean.

The atmospheric forcing fields include 10 m surface wind speed, 2 m temperature, relative humidity, precipitation, downward longwave and shortwave radiations. Through the two-way coupled process between ice and sea, the ocean component model provides the sea ice component model with information such as ablation/freezing potential, SST and salinity, surface velocity, while the sea ice model provides the information of the SIC, fresh water and salinity fluxes, ice-sea

stress and others. The heat flux on the sea ice surface is referred to the results of Parkinson and Washington (1979). The change of sub-grid ice thickness in the sea ice model is taken into account in the calculation of conduction heat flux. In other words, the sea ice is divided into 7 categories according to the SIT value in a horizontal grid. The variation of heat flux and albedo caused by snow cover and the process of snow-ice conversion are considered on the surface of sea ice. The albedo of dry ice, wet ice, dry snow and wet snow is 0.87, 0.78, 0.98 and 0.80, the sea-ice drag coefficient is 5.2, and the sea-ice

intensity is $2.7 \times 10^4$ Pa (Losch et al., 2010).



### 3.2.2 The SMRF data assimilation method

Xie (2005) shows that in the process of data assimilation, if the error of long wavelength could not be well corrected, the short wavelength error could not be well corrected either. The SMRF is a method that can realize the sequential correction from long wavelength to short wavelength and extract multiscale information through a single three-dimensional variational

(3DVAR) analysis (Zhang et al., 2020).

In SMRF, firstly, a recursive filter operator $\mathbf{B}$ with the small filter parameter $\beta$ is applied to the initial guess field. The parameter $\beta$ is set to small value in order to ensure that information of all spatial scales could pass the filter. The cost function and its gradient are calculated based on the filtered initial guess field. Then, another recursive filter operator $\mathbf{E}$ with the parameter $\alpha$ is applied to the negative gradient of the cost function. This filter parameter $\alpha$ should select a larger value

at the beginning to extract the "longest" wavelength information in the observational data. A line search process (More and Thuente, 1994) is performed along this gradient direction to find the appropriate step size, and the estimated value is updated. The observational residual is obtained by removing the extracted signal from the observation data. Then, the filter parameter is appropriately reduced to extract the "maximum" scale signal in the observational residual at current iteration. As the number of iterations increases, the filter parameter $\alpha$ sequentially decreases, so that the information of each scale can be

extracted successively from long wavelength to short wavelength in order to obtain the analysed field.

### 3.3 Evaluation criteria

In order to evaluate the accuracy of the BRMT constructed by the proposed regression model and prove its own superiority by comparison with other data sets (Sect. 4) or other numerical prediction experiments (Sect. 5), five statistical metrics are used in this paper. It includes RMSE, mean deviation (Bias), Pearson correlation coefficient (Coff), standard deviation (STD)

and centred root mean square error (CRMSD). The meaning and specific calculation methods are as follows:

$$\mathrm{RMSE} = \sqrt{\frac{\sum_{i=1}^{n}(y_i - x_i)^2}{n}} \tag{1}$$

$$\mathrm{Bias} = \frac{\sum_{i=1}^{n}(y_i - x_i)}{n} \tag{2}$$

$$\mathrm{Coff} = \frac{\sum_{i=1}^{n}(x_i - \bar{x})(y_i - \bar{y})}{\sqrt{\sum_{i=1}^{n}(x_i - \bar{x})^2}\sqrt{\sum_{i=1}^{n}(y_i - \bar{y})^2}} \tag{3}$$



$$\text{STD} = \sqrt{\frac{\sum_{i=1}^{n}(z_i - \bar{z})^2}{n}} \tag{4}$$

$$\text{CRMSD} = \sqrt{\frac{\sum_{i=1}^{n}\left[(y_i - \bar{y}) - (x_i - \bar{x})\right]^2}{n}} \tag{5}$$

where, $i$ represents the label of point, $x_i$ represents the observed value of the $i$-th point, $y_i$ represents the data set value (or forecast value) of the $i$-th point, $\bar{x}$ and $\bar{y}$ represents the average value, $n$ represents the total number of points with valid values (except land points), $z_i$ and $\bar{z}$ indicate that they can represent both observed values and data set values.

STD, CRMSD and Coff are usually expressed on a polar graph based on the cosine relationship of Eq. (6), which is called Taylor graph. The standardized Taylor chart needs to divide the STD and CRMSD of the observed value and the data set value by the STD of the observed value. The standardized STD and CRMSD of data set are expressed as NSTD and NCRMSD, respectively.

$$\text{CRMSD}^2 = \text{STD}_{obs}^2 + \text{STD}_{dat}^2 - 2\,\text{STD}_{obs} \cdot \text{STD}_{dat} \cdot \text{Coff} \tag{6}$$

$$\text{NSTD} = \frac{\text{STD}_{dat}}{\text{STD}_{obs}} \tag{7}$$

$$\text{NCRMSD} = \frac{\text{CRMSD}}{\text{STD}_{obs}} \tag{8}$$

$\text{STD}_{obs}$ represents the STD of the observed value, $\text{STD}_{dat}$ represents the STD of the data set value.

## 4 Comparison of BRMT with in situ observations and other data sets

### 4.1 Comparison with other data sets

Arctic sea ice has significant seasonal variation. The sea ice extent reaches its maximum from February to March. With the arrival of the melting season, the sea ice decreases rapidly in July, and the melting speed slows down to some extent in August. The minimum sea ice extent is usually reached in September. We qualitatively and quantitatively compare the averaged SIT via the BRMT with the counterparts of three data sets (CMST, PIOAMS and GIOMAS) in July, August and September, respectively.

From 2011 to 2015, the averaged SIT in July of BRMT is below 1.5 m in the Barents Sea, Kara Sea, Laptev Sea, East Siberia Sea and Chukchi Sea. The SIT near the Centre Arctic is about 1.5-2 m, which is relatively consistent with the performance of PIOMAS (Fig. 3a). The SIT from BRMT in the north of the Canadian Arctic Archipelago (CAA) is slightly





lower than that of other data sets, where is mostly covered by ice more than 2.5 m, along with the thick ice more than 3.5 m in the coastal area. In contrast, the SIT from CMST in the Central Arctic and the north of the CAA is about 0.5 m thicker than BRMT on average, and the SIT in other regions is equivalent to BRMT. Different from other data sets, GIOMAS

presents the distribution of thicker ice thickness in the whole region, especially for the depiction of thin ice below 1 m.

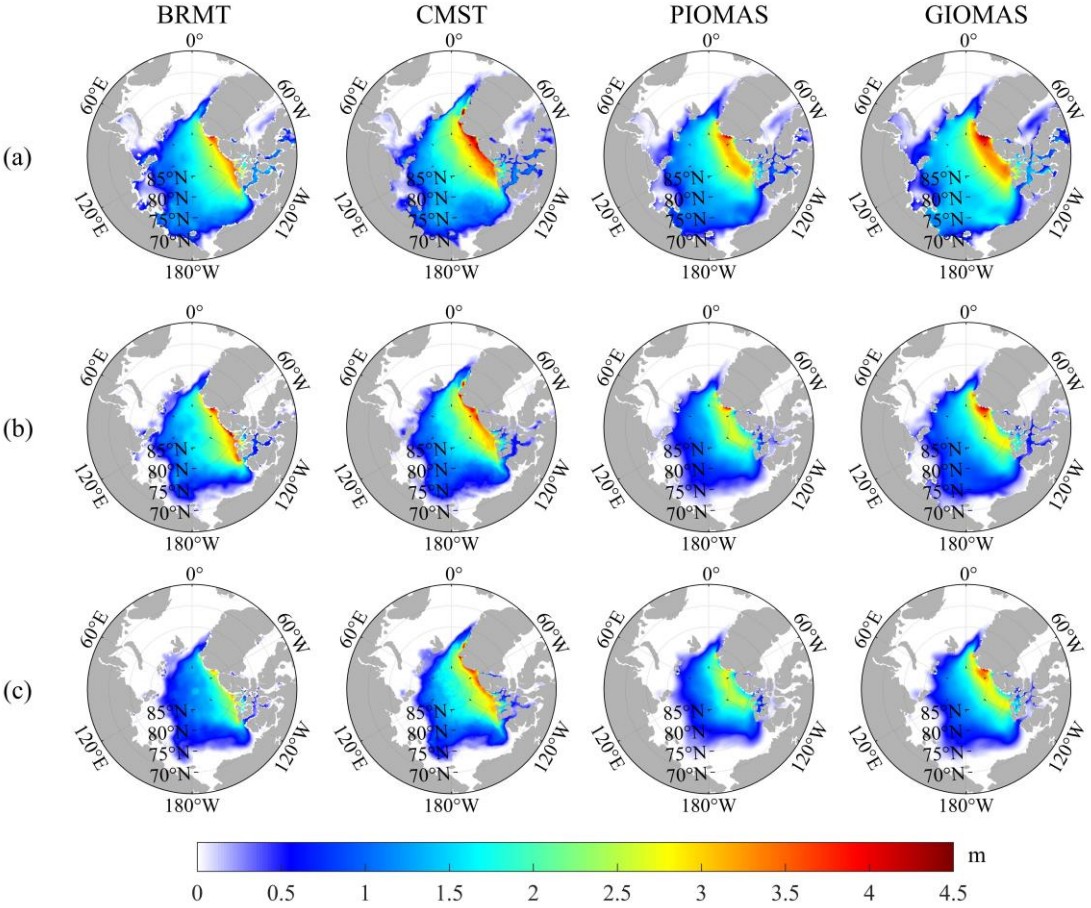

**Figure 3.** Spatial distribution of sea ice thickness (m) in July (a), August (b) and September (c) averaged from 2011 to 2015 for BRMT, CMST, PIOMAS and GIOMAS.

The differences of the SIT distribution between the four data sets are similar in August and September (Figs. 3b-c). It is

worth mentioning that PIOMAS always seems to tend to have a smaller sea ice extent and it is difficult to reproduce the thick ice along the coast, but BRMT have a larger sea ice extent and the meridional gradients at a narrow band along the northern coast of Greenland and the CAA is much steeper in the BRMT than in the PIOMAS. In addition, the ice thickness around the poles of CMST in the whole melting season is always higher than that of other data sets and even in September, the ice thickness is at least about 1.75 m. In contrast, the SIT in September of BRMT around the pole is lower, about 1.25 m.

In fact, in the detailed comparison between the data sets and the SIT observation of IMB in Sect. 4.2.3, it can be seen that





CMST does generally overestimate the SIT in the central Arctic, and the SIT from BRMT corrects some of these biases present in CMST.

The frequency distribution of the SIT from BRMT in different ice thickness ranges and different months of melting season is almost consistent with the other three data sets (Fig. 4). Overall, in the melting season, 0-1 m thin ice accounts for about half

of the total ice thickness classification, which is the main ice thickness range. The frequency of the SIT from BRMT in this range is higher than that of CMST and GIOMAS, increasing by about 12.24 % and 22.45 %, respectively. Correspondingly, in the thick ice above 3.5 m, which accounts for less than 1 % of the total ice thickness classification, the frequency of the SIT from BRMT is significantly lower than CMST and GIOMAS by 71.86 % and 62.09 %, respectively. Although there is not enough in situ SIT observation data covering the whole Arctic in the melting season to prove which data set is more in

line with the real situation, previous studies have shown that the SIT simulated by GIOMAS in spring and autumn is always thicker than that observed by cryosat-2 (Watanabe et al., 2019). In addition, the difference between PIOMAS and ICESat from February to March in 2004-2008 indicates that PIOMAS has too much thin ice (Schweiger et al., 2011). Compared with PIOMAS, which consistently have the maximum frequency in the thickness range of 0-0.5 m throughout the melting season, the frequency of the SIT from BRMT in this range decreased by 9.63 %.

From the discussion above, in the melting season, the SIT from BRMT will not overestimate the SIT below 0.5 m, nor will there be a large extent of thick ice. It is a better compromise among several other SIT data sets. This fully reflects the advantages of big data and machine learning, that is to say, BRMT can make full use of the sea ice data and real-time satellite observation data for many years in history, so as to obtain higher precision ice thickness.

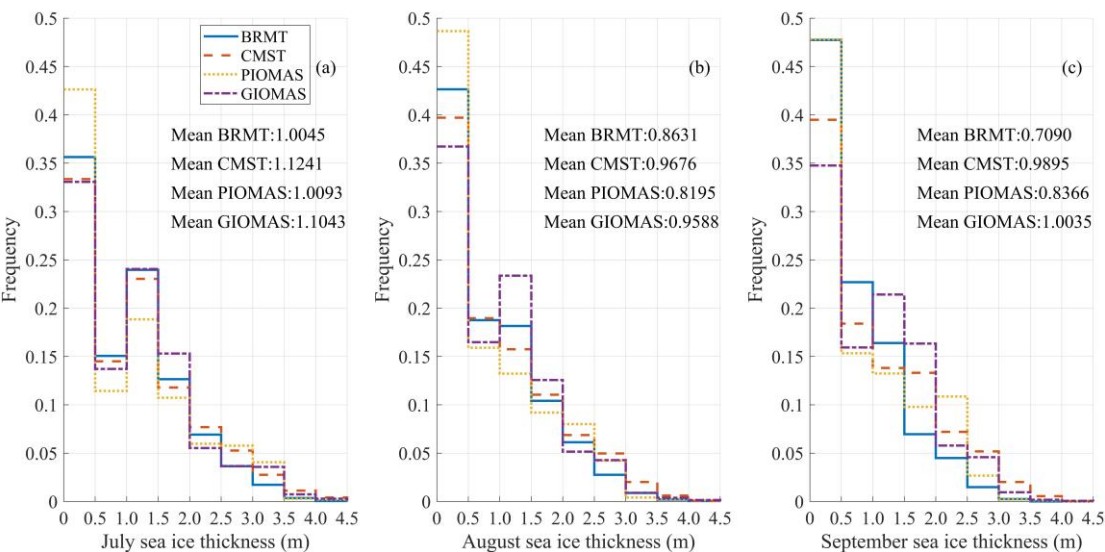

**Figure 4.** Histograms of mean sea ice thickness (m) frequency distributions in July (a), August (b) and September (c) averaged from 2011 to 2015 for BRMT, CMST, PIOMAS and GIOMAS.




## 4.2 Comparison with BGEP ULS data

Through the analysis in Sect. 4.1, we know that the SIT of GIOMAS is similar to CMST in the frequency distribution of thin ice and thick ice, and GIOMAS has biases that is harder to ignore compared with the observation. Therefore, considering the simplicity of description and the representativeness of data sets, only three SIT data sets from BRMT, CMST and PIOMAS are retained when compared with BGEP mooring equipment and IMB buoy. In addition, in order to facilitate the comparison with the in situ observation, the objective analysis method is used to process the SIT of data sets (BRMT, CMST and PIOMAS in this section or forecast model in Sect. 5) at grid points in a certain range near the observation points. The calculation formula is as follows:

$$Z_{ij}^a = \frac{\sum_k w_k z_k^o}{\sum_k w_k} \tag{9}$$

where, $k$ is the number of the data set grid points within the influence radius of the observation position, $z_k^o$ represents the SIT at the $k$-th data set grid point, $Z_{ij}^a$ represents the averaged SIT of the data set within the influence radius of the observation position, and $w_k$ represents the weight. The calculation formula is as follows:

$$\begin{aligned} w_k &= e^{\frac{r_k^2}{2R^2}}, & r_k &\leqslant R \\ w_k &= 0, & r_k &> R \end{aligned} \tag{10}$$

where, $R$ represents the influence radius, and its value is determined by the density of data set grid points near the observation position. $r_k$ represents the distance between the $k$-th data set grid point and the observation point.

In order to make a visual comparison with the SIT observations at three mooring locations of BGEP (BGEP_A, BGEP_B and BGEP_D), the time series diagrams between the SIT of BRMT, CMST, PIOMAS and the observations at different locations are drawn respectively (Fig. 5). It is not difficult to find that the three data sets basically reproduce the changes of SIT in different years and months. Compared with CMST and PIOMAS, which depend on the SIT of the background field based on numerical model simulation, the SIT from BRMT constructed based on the SIC of the single grid point has stronger fluctuation in a short time, which is more similar to observation. The numerical model is affected by factors such as resolution, meteorological forcing and the completeness of dynamic equation, which is difficult to reproduce for some small-scale processes. At the same time, the BRMT model is separately operated for each grid every day, resulting in the independent of the constructed SIT in adjacent time and space, which can describe some nonlinear processes and reproduce some small-scale disturbances and abrupt change. It also makes the SIT from BRMT more predictable for the overall variation trend of SIT increasing or decreasing than the other two data sets with more gentle variation. For instance, in September 2014, at the position of BGEP_D mooring equipment, the SIT firstly decreases and then increases slightly. The





SIT from BRMT well captures this feature. In contrast, the change range of PIOMAS is very weak, and CMST even shows a
trend of rising first and then falling (Fig. 5c).

Except for 2013 and 2014 at BGEP_B station, the SIT from BRMT is closer to the observed data than the CMST and
PIOMAS data as a whole (Fig. 5). In July every year, the SIT at each station exceeds 1m, and then decreases rapidly with
time; The sea ice thickness reaches the minimum at the end of August and the beginning of September, and can be reduced
to zero in some years; After mid-September, the sea ice thickness gradually increased. The SIT from BRMT is larger than
that from observation and CMST and PIOMAS data during 2013 and 2014 at BGEP_B station. In order to further analyse
the reason for the error of the SIT from BRMT, the reanalysis data set TOPAZ used to build BRMT is added to the
comparison in Fig. 5, as shown by the yellow dotted line. We note that TOPAZ has significantly overestimated SIT in most
time periods compared to the observations of the three mooring facilities so the SIT from BRMT is inevitably affected by the
positive deviation of the basic data set. At the same time, an obvious conclusion is that the SIT processed by BRMT has
better consistency with the observations than TOPAZ. This can be attributed to the fact that the relationship between SIC and
SIT in TOPAZ is relatively real under the constraints of sea ice physical equation in the numerical model. Based on this
relationship, the SIC from satellite remote sensing is used by BRMT to construct SIT, so that it is more accurate than
TOPAZ.

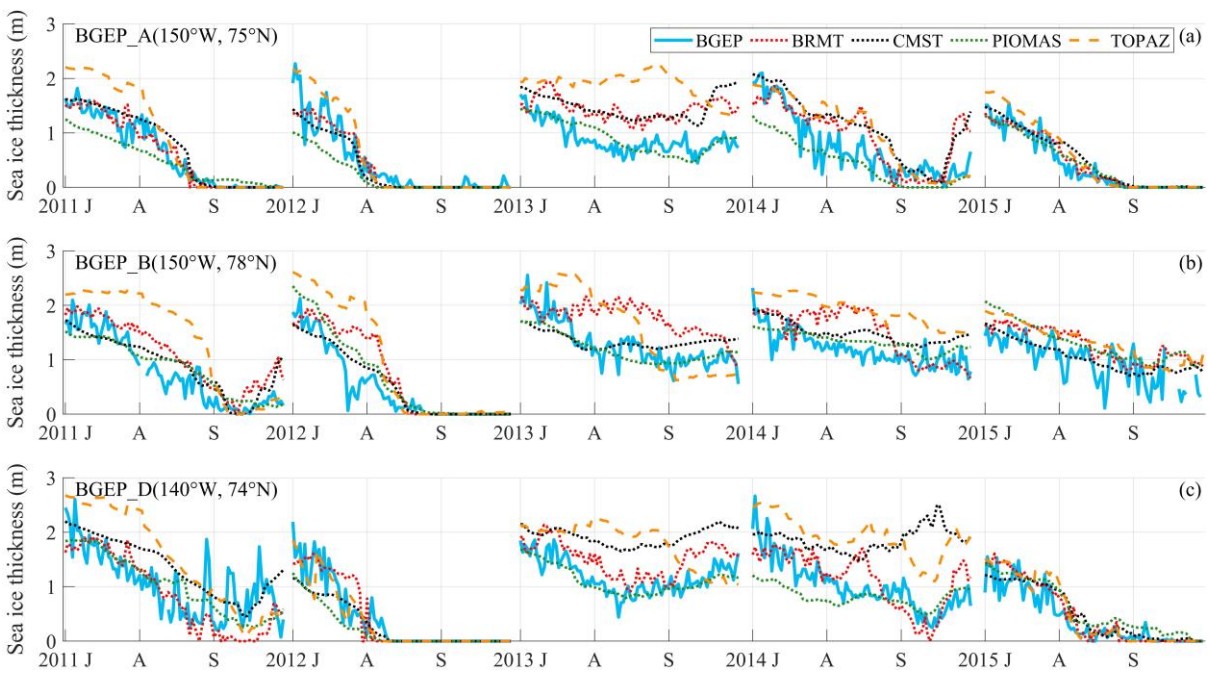



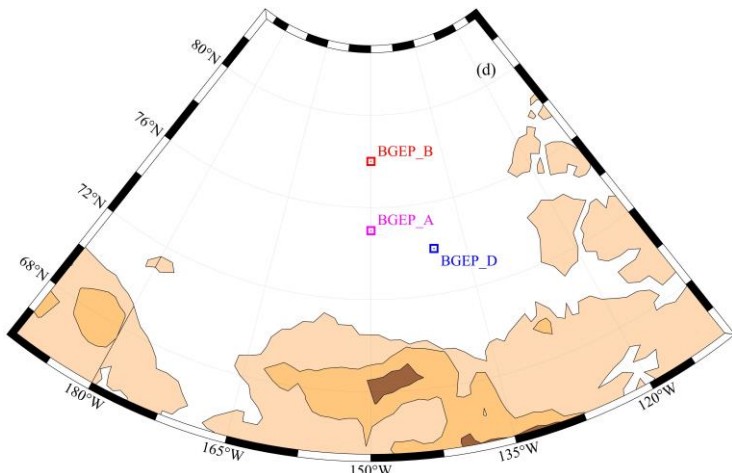


**Figure 5.** Time series of sea ice thickness (m) during melting season from 2011 to 2015 for BGEP ULS data (blue), BRMT (red), CMST (black), PIOMAS (green) and TOPAZ (yellow) at BGEP mooring facilities BGEP_A (a), BGEP_B (b), and BGEP_D (c). J, A and S represents July, August, and September, respectively. Locations (d) of mooring facilities BGEP_A (150° W, 75° N), BGEP_B (150° W, 78° N), and BGEP_D (140° W, 74° N) are represented by magenta box, red box, and blue box, respectively.

Then, Fig. 6 displays the scatter diagrams of ice thickness between three data sets (BRMT, CMST, PIOMAS) and three mooring facilities (BGEP_A, BGEP_B, BGEP_D). Compared with the in situ observation of mooring equipment at BGEP_D location, CMST obviously tends to overestimate the thin ice below 1 m. The simulation bias of CMST for ice thickness about 0.5 m is even as high as 2 m, and the correlation with observation is only 0.7544. In contrast, the correlation of the SIT from BRMT is better than the other two data sets (R=0.843), and the average bias is only 0.0675 m. From the SIT

time series of BGEP_D in Fig. 5, it can be seen that the maximum deviation between CMST and observation always occurs in September. In fact, the geographical location of BGEP_D in September is located at the junction of outer thin ice area and inner thick ice area of sea ice in Beaufort Sea (Fig. 3c). When the SIT from BRMT is still in the stage of sea ice melting in September as observed, the SIT of CMST has gradually begun to freeze, resulting to the thicker SIT compared with the observation. This may be due to the fact that CMST has no SIT data for assimilation in the melting season, resulting in that

the SIC and SIT in the transition area between thin ice and thick ice are not well related in essence, while BRMT makes up for the defect in the matching range between SIC and SIT.

On the whole, PIOMAS has better correlation with the observation of three mooring facilities. However, compared with the SIT from both BRMT and CMST, it always tends to underestimate the SIT with more than 1.5 m, especially at the beginning of the melting season (such as July in 2011-2014 at BGEP_A location, July in 2012 and 2014 at BGEP_D location). In

addition, according to the calculation, the deviation of SIT between the TOPAZ and the observation at BGEP_B is the largest among the three mooring facilities (Fig. 6k). This may be the most likely reason why the SIT from BRMT performs slightly worse at BGEP_B than CMST and PIOMAS (middle column of Fig. 6).



**Figure 6.** Scatter plot of sea ice thickness (m) during 2011-2015 melting season between BRMT (a-c), CMST (d-f), PIOMAS (g-i),
TOPAZ (j-l) and BGEP moorings facilities BGEP_A (a, d, g, j), BGEP_B (b, e, h, k), BGEP_D (c, f, i, l), respectively. The blue line
indicates equality and the red line represents the best fit to the observations. The number of total observation points, data set bias, and
dataset-observation correlation (R) are listed.

The qualitative comparison of error evaluation criteria among the SIT from BRMT, CMST and PIOMAS is further
quantified. The details of RMSE and Bias in each year are shown in Fig. 7. We divided three mooring facilities and five
years into 15 groups of data, each of which contains one for BRMT, one for CMST and one for PIOMAS. Based on this,
according to the comprehensive performance of RMSE and Bias, it is statistically obtained that in 8 groups, the SIT from
BRMT is significantly better than CMST and PIOMAS or not obviously different from the results of an optimal data set, of
which there are 4 groups in each of BGEP_A and BGEP_D. For the 4 groups in BGEP_B, the SIT of BRMT is worse than
the other two data sets, with the worst result in 2013 (RMSE=0.6124 m, Bias=0.4862 m). The RMSE and Bias of CMST are
apparently greater than those of BRMT and PIOMAS in BGEP_D and they even reach 0.9888 m and 0.8270 m, respectively





in 2014. The Bias of PIOMAS SIT has more negative values, which further indicates that the SIT of PIOMAS is underestimated relative to observation.

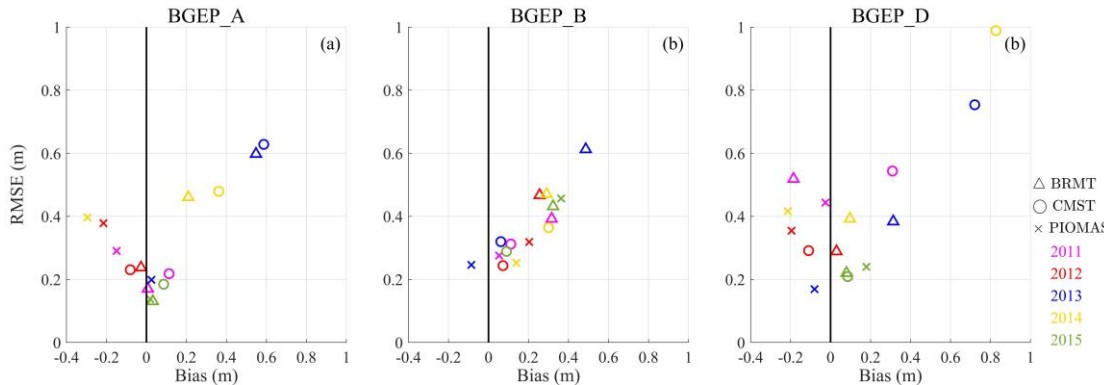

**Figure 7.** RMSE (m) versus bias during melting season from 2011 to 2015 (marked in different colours) for BRMT (△), CMST (○) and PIOMAS (✕) relative to BGEP moorings BGEP_A (a), BGEP_B (b) and BGEP_D (c).

Based on the above analysis, we attempt to give some conclusive suggestions on the selection of SIT data sets in the melting season. For the region near the location of BGEP_A and BGEP_D mooring facilities, BRMT is not a disappointing option. At BGEP_B, however, BRMT is not recommended, and CMST may be the better choice. It should be noted that near the location of BGEP_D, the CMST is not convincing. Compared with the observation, BRMT and CMST are more likely to overestimate SIT, while PIOMAS is more likely to underestimate SIT.

## 4.3 Comparison with IMB buoy data

Compared with the fixed position BGEP mooring equipment, the IMB buoy data can provide us with more information about the temporal and spatial variation of SIT. For comparison, a total of 25 available buoy data in the melting season from 2011 to 2015 are selected (Fig. 8d). Because Taylor diagram can illustrate how well the models match in terms of correlation, RMSE and the ratio of variances (Taylor, 2001), it is considered to summarize the relative merits of different models and track changes in performance of models. In addition, there are systematic deviations for CMST and PIOMAS, and they can be reduced by removing the average thickness of each data set (Mu et al., 2018a). Accordingly, CRMSD and STD of SIT in BRMT, CMST and PIOMAS are normalized. The Taylor diagrams of the three data sets are shown in Figs. 8a-c respectively.

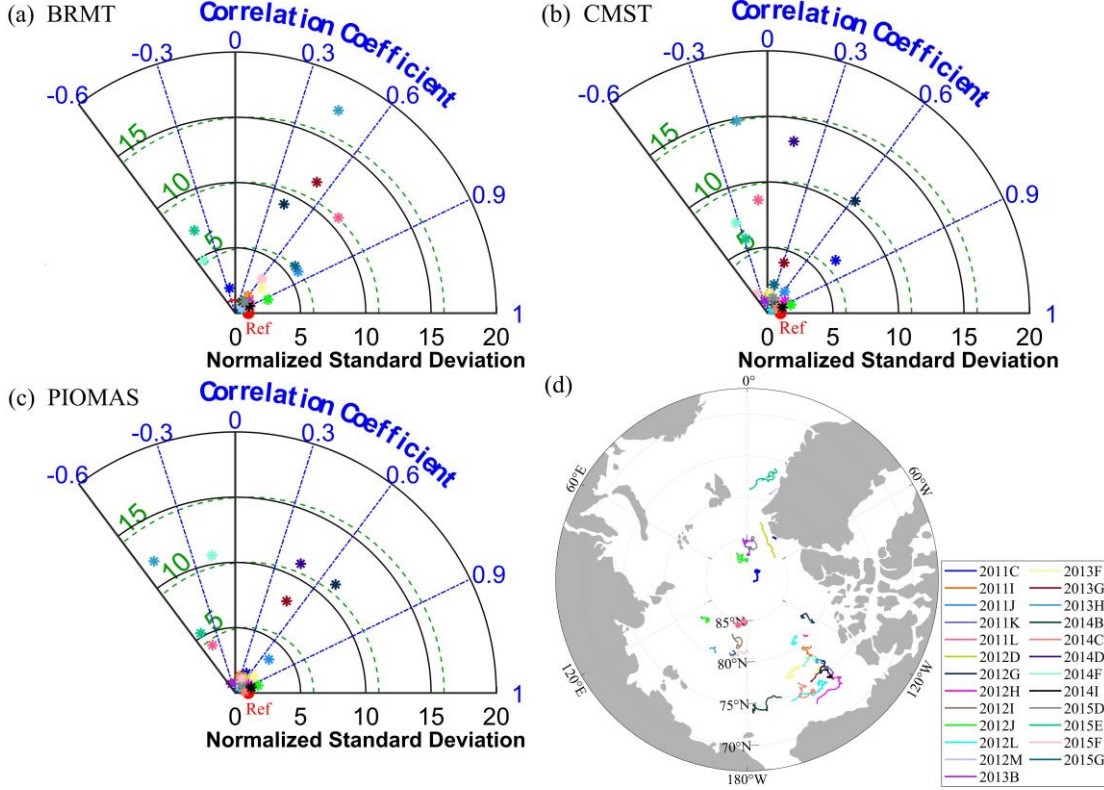

**Figure 8.** Taylor diagrams of (a) BRMT, (b) CMST and (c) PIOMAS with respect to all available IMB buoy data during melting seasons from 2011 to 2015. The green dotted lines indicate the normalized CRMSD. The trajectories of all the IMB buoys are shown in (d). The reference observations are indicated by Ref in red.

Generally speaking, most of the points in BRMT, CMST and PIOMAS are concentrated in the range where NSTD and NCRMSD are less than 5, and only some sporadic points are scattered outside. We make an in-depth analysis of the points of these two parts. In the part with small deviation evaluation criteria value (Enlarged figure is not shown), the correlation between the SIT from BRMT and IMB buoy data is better. The averaged value of correlation coefficient in the SIT from BRMT is about 0.69, while in CMST and PIOMAS, this value is 0.46 and 0.48 respectively. The CRMSD of the SIT from BRMT is not obviously different from CMST and PIOMAS, and the normalized values of three data sets are 1.40, 1.34 and 1.38 respectively. However, compared with the NSTDs of CMST and PIOMAS (1.41 and 1.52), BRMT performed slightly worse (1.80). The changes of different models on different statistical variables can be explained. Because the construction of BRMT is independent in space and time, it has no constraints or correlation, which leads to the fluctuation of SIT in adjacent space and time is greater than that in other data sets. Nevertheless, this also explains why the difference in NCRMSD, which removes the mean of time series (Mu et al., 2018a), between BRMT and the other two data sets is not as large as that in NSTD. What's more, the better correlation between the SIT from BRMT and IMB buoy maybe is related that BRMT





375 contains more historical information of SIT in melting season, which also means that the SIT from BRMT can better reflect
the variation trend than CMST and PIOMAS.

It can be clearly seen from Figs. 8a-c that the SIT of the three data sets collectively shows a large deviation at the same six
IMB buoys, namely 2011L, 2012G, 2013G, 2013H, 2014F and 2015E. Actually, the SIT of the six IMB buoys show the
same characteristics. Here, the IMB_2013H buoy (Fig. 9a) is chosen as an example to illustrate because its NSTD and

380 NCRMSD in the three data sets reached the maximum. The SIT of this buoy has almost no change from 3-30 September,
2013 and the STD is only 0.009 but the SIT from BRMT, CMST and PIOMAS show some fluctuations. Although the
fluctuation amplitude is not violent, the ratio of STD and CRMSD in three data sets to the observed STD is more than 10
times.

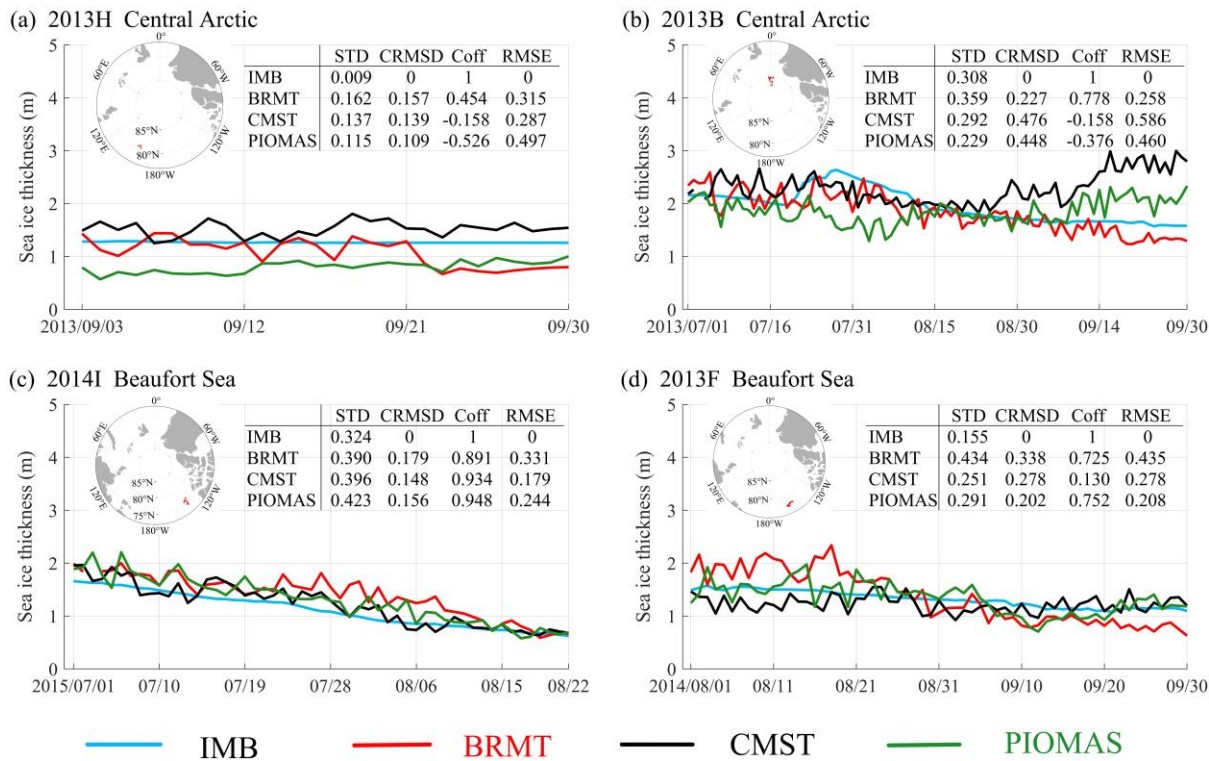

**Figure 9.** Sea ice thickness (m) time series on the trajectories of (a) 2013H, (b) 2013B, (c) 2014I and (d) 2013F in different region: IMB
buoy data (blue), BRMT (red), CMST (black), PIOMAS (green) (The date format is yyyy/mm/dd). Each IMB buoy trajectory is shown in
the top left corner which is indicated by a red line. The statistics for IMB buoy data, BRMT, CMST, and PIOMAS are also shown in each
plot (STDs and CRMSDs are unnormalized).

Based on the above analysis, in order to analyse the results more comprehensively and deeply, an additional comparison of
RMSE between the SIT from BRMT, CMST, PIOMAS and IMB buoys is added (the complete results can be seen in Table
A2 in the Appendix). In order to enhance the readability of the results, the RMSEs of the SIT from BRMT, CMST and
PIOMAS are divided into three levels: excellent, good and failed. Then, the RMSEs in Table A2 are classified according to





error levels, sea ice types and regions. The number of RMSEs in each category is counted for BRMT, CMST and PIOMAS, respectively, just as shown in Fig. 10. First of all, in terms of the division of regions, there are the most buoys in the Central

Arctic and Beaufort Sea, with 11 buoys. Because the trajectory of the only one buoy in the Chukchi Sea during the study period is in the Central Arctic, its results are counted into the Central Arctic for the convenience of expression. In the Central Arctic, the overall performance of the SIT from BRMT is extremely better than that of CMST and PIOMAS, and the RMSE of only one buoy is the worst in the three data sets (Fig. 10a). In the Beaufort Sea, although the number of BRMT in the excellent level is not a few, the number of BRMT in the failed level is also large, and the overall result is basically the same

as that of PIOMAS (Fig. 10b). There are relatively few buoys in E Greenland Sea and Laptev Sea, only 2 and 1 respectively. There is a remarkable difference in the performance of BRMT in these two regions. Compared with CMST and PIOMAS, the results are the best in E Greenland Sea and the worst in Laptev Sea (Figs. 10c-d).

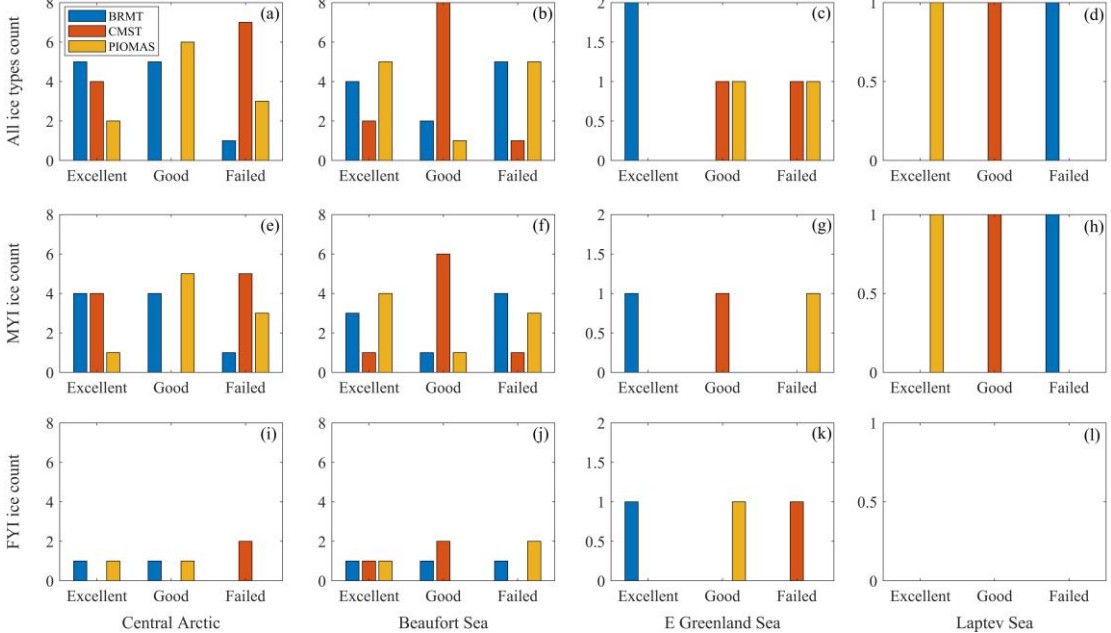

**Figure 10.** Bar chart of total numbers for BRMT (blue), CMST (red) and PIOMAS (yellow) distributed in three scales (excellent, good,

faired) based on RMSE metric statistics of all datasets, respectively. We show values over all ice types (a-d), MYI (e-h), FYI (i-l) and over Central Arctic (a, e, i), Beaufort Sea (b, f, j), E Greenland Sea (c, g, k), Laptev Sea (d, h, l). MYI=Multi-year Ice, FYI=First-year ice.

Now the representative time series of SIT are discussed respectively (Figs. 9b-d). The IMB_2013B buoy located in the Central Arctic, shown in Fig. 9b, has a complete record of the SIT throughout the melting season in 2013. It can be seen from the figure that the SIT from BRMT perfectly captures the trend that the SIT firstly thickens slightly and then decreases

gradually, and the correlation coefficient with the observation reaches 0.78. On the contrary, CMST and PIOMAS go the opposite way. In the time series of other buoys in the Central Arctic, CMST almost always tends to overestimate SIT, while PIOMAS tends to underestimate SIT most of the time. Especially in September at the end of the melting season, the SIT of



CMST invariably tends to increase during this period, but in fact, the sea ice continues to slowly melt. Nevertheless, the SIT from BRMT captures the variation of SIT almost every time. It is inferred that the BRMT with the historical correlation
information between SIC and SIT can better reflect the change trend of SIT. It is gratifying that BRMT not only performs well in multi-year ice (MYI), but also shows quite good performance in first-year ice (FYI) (Figs. 10e, i). It is remarkable that due to the limited horizontal resolution of the basic data used to construct SIT, the SIT error of BRMT near the land is also relatively large.

As shown in Fig. 9c, the observed SIT in the Beaufort Sea continues to decline from 1.3 m in mid-late July to 0.8 m in early-
August, while the averaged SIT of BRMT remain at 1.5 m in mid and late July, showing a downward trend after August 6, and then close to the observation in mid-August. In September of Fig. 9d, the SIT has been slightly underestimated by BRMT. Especially at the end of September, the SIT of BRMT still fluctuated up and down at 0.8 m, while the observed SIT was always above 1.1 m. These are speculated that BRMT has the lag response to the initial melting state and the stagnation to maintain the melting state at the end of melting season when facing the regions with the rapid melting or freezing of sea
ice. Although the performance of the SIT from BRMT in the Beaufort Sea is not as prominent as that in the Central Arctic, some regularities in different periods are still found, which also provides some new ideas for the improvement of the subsequent model.

Compared with the Central Arctic and the Beaufort Sea, the number of buoys in the E Greenland Sea and the Laptev Sea is too small to draw some convincing conclusions. However, we are gratified that the SIT from BRMT has a brilliant
performance in both MYI and FYI in the E Greenland Sea (Figs. 10g, k). Finally, the performance of the SIT from BRMT in different ice types has no significant tendency to be good or bad, so it can be considered that the above conclusion is valid for all ice types. This conclusion may be more reliable if more buoys of the type of FYI are involved in the comparison.

## 5 Retroactive real-time forecast experiments

In the Sect. 4, we have made a detailed analysis of the advantages and disadvantages of the reconstructed SIT via BRMT
compared with other data sets and its accuracy compared with in situ observation. However, it still makes us wonder whether the forecast accuracy of Arctic sea ice variables can be improved when the real-time reconstructed SIT from BRMT is introduced into the assimilation process? As a consequence, taking September 2011 as an example, based on the SIT from BRMT, this section uses the multi-scale sea ice multi-element joint assimilation method to carry out the retroactive real-time forecast experiments in the Arctic melting season.

### 440   5.1 Experiment design

In the forecast experiments, numerical results of SIC and SIT from the ice-ocean coupled model are used as the background field on 1 September 2011; the SIC from satellite remote sensing and the SIT from BRMT at the corresponding time are used as the observation field (The SIT field from on 1 September is shown in Fig. 11). Using the SMRF method, SIC and





SIT observations are assimilated to update the background field to obtain the analysed one; then, the analysed field is
regarded as the initial forecast field, and the forecast results from 2-8 September 2011 are obtained by the 7 d integration of
the ice-ocean coupled model. In the next step, the 24 h forecast results (i.e., 2 September) are used as the background field
and the observed SIC and the SIT from BRMT at the corresponding time are assimilated to provide the initial field of the
next 7 d forecast; the forecast results from 3-9 September 2011 are obtained by model simulation. According to this process,
the data assimilation and model integration are alternately rolled until 30 September 2011, and the one-month numerical
forecast of Arctic sea ice is realized.

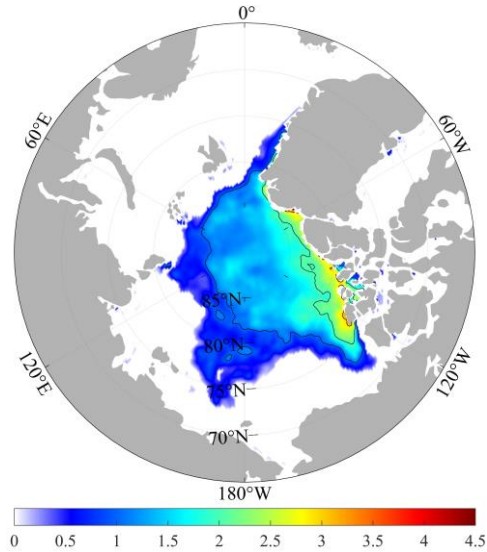

**Figure 11.** The constructed sea ice thickness from BRMT on 1 September 2011 (unit: m).

**Table 1.** Comparison of Exp_Ctrl, Exp_SIC and Exp_SIC&SIT initial fields.

| Experiment title | Assimilation method | Assimilated SIC | Assimilated SIT |
|---|---|---|---|
| Exp_Ctrl | None | None | None |
| Exp_SIC | The SMRF method | SIC from Satellite remote sensing | Empirically adjusted SIT |
| Exp_SIC&SIT | The SMRF method | SIC from Satellite remote sensing | The SIT from BRMT |

The initial fields for the three experiments are shown in Table 1. The Exp_Ctrl is a control experiment, which does not
assimilate any data and only integrates forward through the ice-ocean coupled model. The other two experiments both use
the SMRF data assimilation method, but differ in the construction of the initial SIC and SIT. The Exp_SIC only assimilates
the SIC and empirically adjusts the SIT. The Exp_SIC&SIT jointly assimilates both the SIC and the SIT from BRMT.



## 5.2 Results

### 5.2.1 Sea ice concentration forecast

For intuitive and effective comparison, RMSE time series of SIC forecast results relative to satellite remote sensing observation in the three experiments (Exp_Ctrl, Exp_SIC, Exp_SIC&SIT) are shown in Fig. 12a. In order to more clearly express the difference of SIC between joint assimilation and single-variable assimilation, RMSE time series in the Exp_SIC and Exp_SIC&SIT experiments in Fig. 12a are enlarged (Fig. 12b).






**Figure 12.** (a) RMSEs of sea ice concentration during 2 September to 7 October 2011 (each segment represents the 7 d forecast) between the forecast results of Exp_Ctrl (green), Exp_SIC (red), Exp_SIC&SIT (blue) and the SSMI observation. (b) Enlarge the results of Exp_SIC and Exp_SIC&SIT in (a).

It can be seen from Fig. 12a that the RMSE in Exp_Ctrl is within the range of 0.2-0.25, with an average value of 0.23. However, RMSEs in Exp_SIC and Exp_SIC&SIT are in the range of 0.07-0.12, which are much smaller than Exp_Ctrl. In other words, data assimilation greatly reduces the deviation between the forecast results and the satellite observation. It is not difficult to see from Fig. 12b that although the two experiments use the same initial field of SIC, the RMSE in Exp_SIC&SIT is always smaller than that in Exp_SIC no matter in which forecast period. Obviously, the initial fields of SIC and SIT are improved in Exp_SIC&SIT at the same time, which makes the dynamic coordination of SIC and SIT better in the physical meaning, and then indirectly improves the prediction accuracy of SIC.

In addition, the relationship graph between the forecast error and the forecast time suggests that RMSEs of 1-7 days SIC forecast results in Exp_SIC&SIT are significantly smaller than those in Exp_SIC, especially during the period from 2-12 September and 20-27 September (Fig. 13). This indicates that the improvement of the SIT initial field not only significantly improves the SIC forecast accuracy, but also has a long-term stable effect. To sum up, the Exp_SIC&SIT with BRMT added to the assimilation of the original single variable, whether in terms of the performance of the overall RMSE of each experiment, or in terms of the variation of the error with the prediction time, shows remarkable stability and accuracy in the prediction application of Arctic SIC.

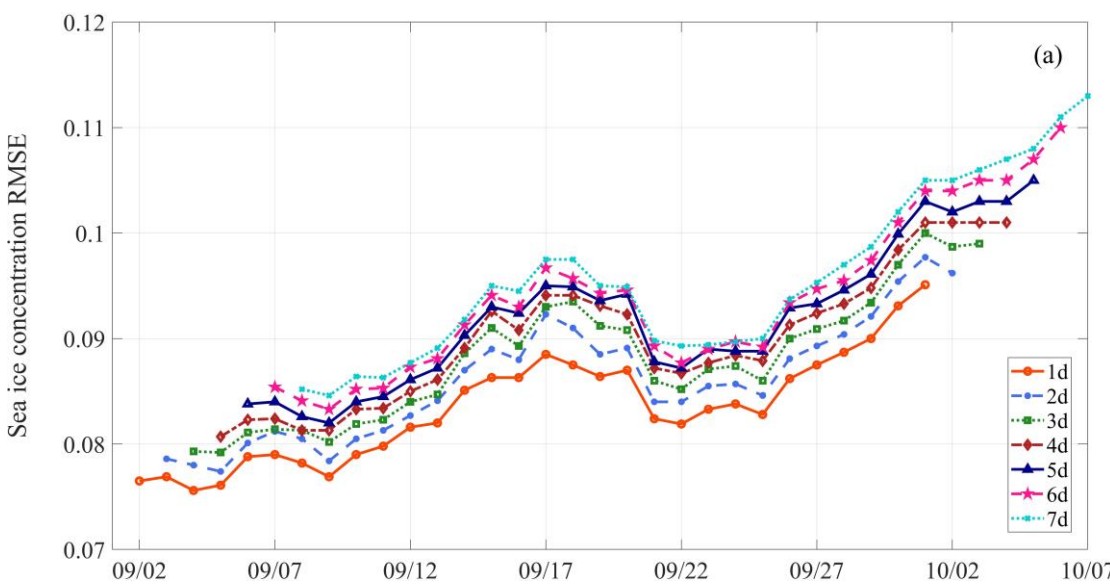





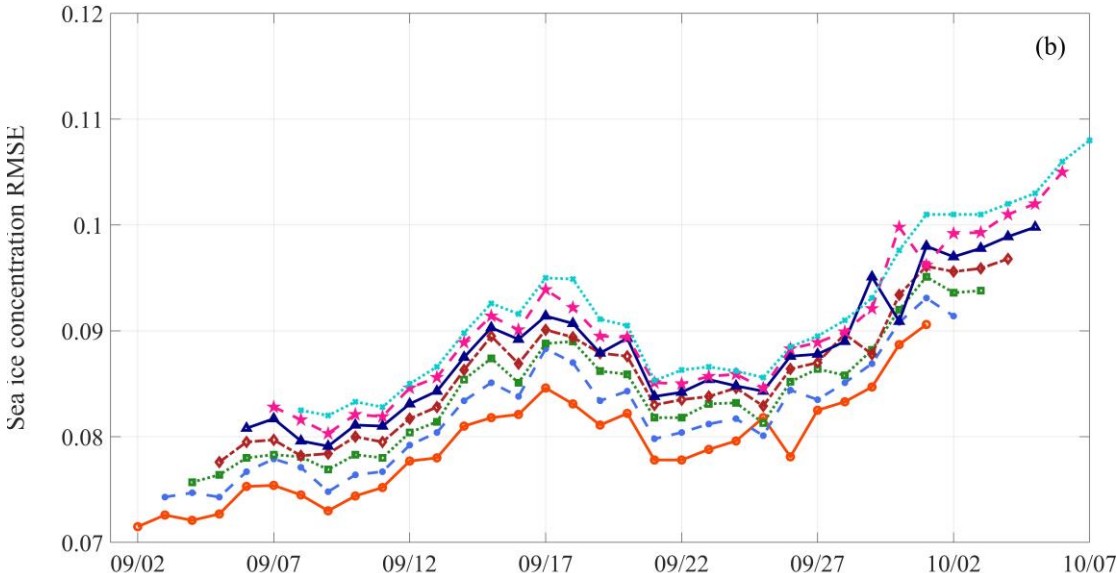

**Figure 13.** RMSEs of the 1 d (orange), 2 d (blue), 3 d (green), 4 d (brown), 5 d (dark blue), 6 d (pink), 7 d (sky blue) sea ice concentration forecast results in Exp_SIC(a), Exp_SIC&SIT (b) relative to the SSMI observation during the period of 2 September to 7 October 2011.

Here, we give a specific example to illustrate the difference of sea ice freezing process between the two cases of assimilating SIC and jointly assimilating both SIC and SIT. The observation indicates that there is a relatively obvious sea ice freezing process during 17-20 September 2011 (Fig. 14a). Although the SIC forecast results from the two comparative experiments starting on 17 September also can capture this process well (Figs. 14b and 14c), the Exp_SIC has a wider range of SIC values greater than 0.7 compared with the observation and Exp_SIC&SIT. For further accurate analysis, the marginal ice zone (defined as SIC < 0.3) of satellite observation and two experiments are plotted respectively, as shown in Fig. 15. As shown in Figs. 14b and 15b, the outline and hierarchical structure of marginal ice zone in Exp_SIC are relatively vague, and especially in the range of 120° W-160° W, the marginal ice zone is fractured or missing. Moreover, with the increase of prediction time, more and more scattered sea ice inconsistent with the observation is generated outside the marginal zone. In Exp_SIC&SIT, which jointly assimilates the SIC and SIT, the extent of the marginal ice zone is basically consistent with the observation, the changes of SIC are well-bedded, and the fine structure of the sea ice in the East Siberian Sea can also be accurately captured. The essential reason for the obvious difference in the marginal ice zone between the two comparative experiments is that the extent of SIC and SIT in Exp_SIC&SIT can match each other, and the corresponding relationship between them is reasonable. However, for Exp_SIC, driven by the model background field of SIT with large error, the sea ice in the SIT forecast field is too thick, and even the SIT at the marginal zone is almost the same as that in the inner zone. These reasons indirectly lead to the narrow extent of marginal ice zone in its SIC forecast field, which can not precisely reflect the real state of sea ice. The same conclusion can be obtained in other forecast periods.



**Figure 14.** Sea ice concentration (b, c) forecasts of Exp_SIC (b) and Exp_SIC&SIT (c) for 1-4 days based on 17 September 2011. The observed sea ice concentration fields (a) on the corresponding day are shown.


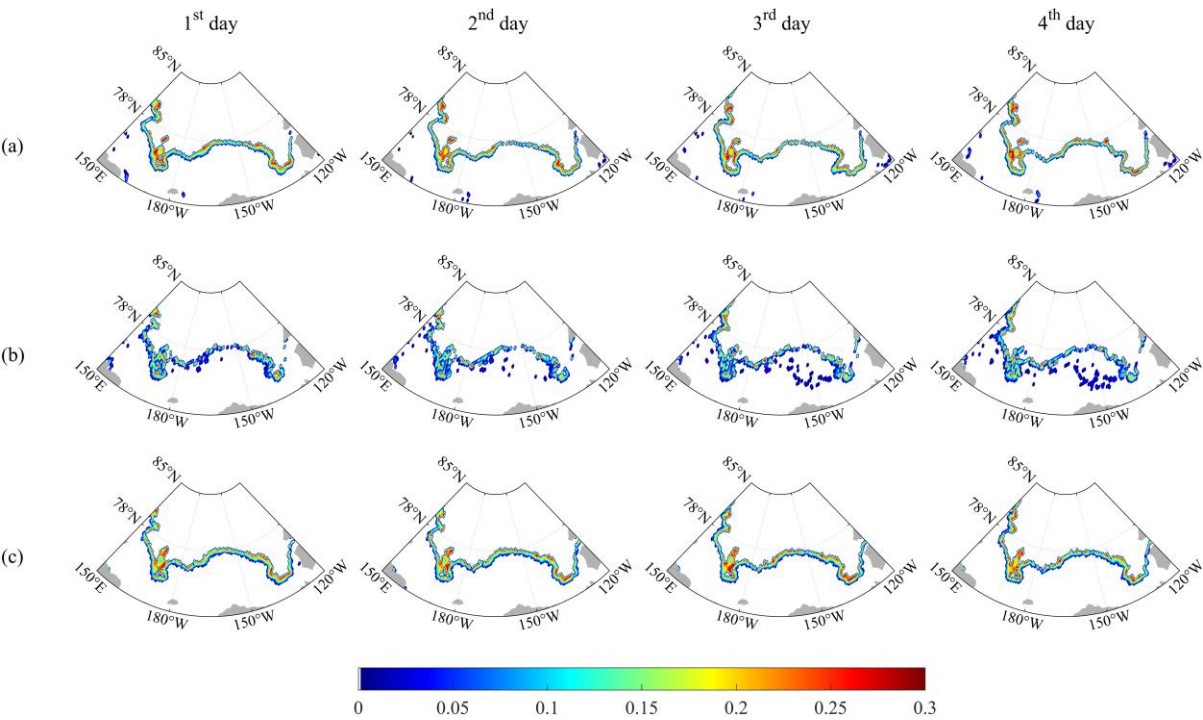

**Figure 15.** Snapshots of marginal ice zone forecasts for 1-4 days on 17 September 2011. The observation (a) and the forecasts of Exp_SIC (b) and Exp_SIC&SIT (c) are shown.

### 5.2.2 Sea ice thickness forecast

Figure 16 compares the 24 h forecast SIT of Exp_Ctrl, Exp_SIC, and Exp_SIC&SIT with the observed SIT from ULS mooring facilities (BGEP_2011B, BGEP_2011D) and buoys (IMB_2011K, IMB_2011L).

It is not difficult to see from Fig. 16 that the predicted SIT of Exp_Ctrl continuously tends to be overestimated. Exp_Ctrl has a flat performance when the observation fluctuates sharply with time, while Exp_Ctrl shows an abnormal fluctuation when the observation is flat. It can be seen from Table 2 that the predicted SIT of the other two experiments with the data assimilation (Exp_SIC and Exp_SIC&SIT) are more precise than that of Exp_Ctrl without data assimilation.

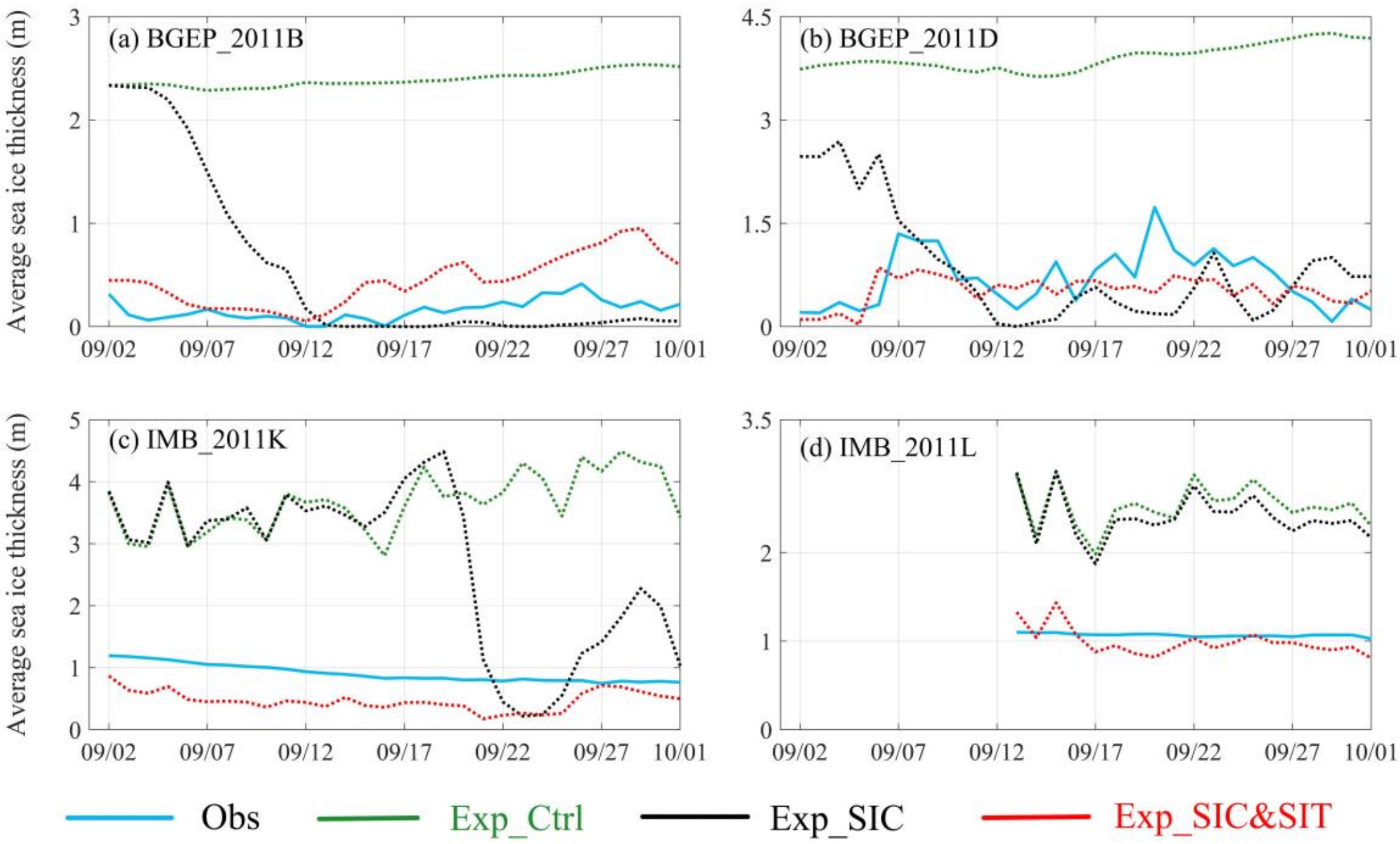

**Figure 16.** Comparison of averaged sea ice thickness (unit: m) from 1-30 September 2011 between the 24 h forecast results of Exp_Ctrl (green dotted line), Exp_SIC (black dotted line) and Exp_SIC&SIT (red dotted line) and the observations (solid blue line) of BGEP_2011B (a), BGEP_2011D (b), IMB_2011K (c), IMB_2011L (d).

**Table 2.** Average absolute deviations of sea ice thickness (m) between 24 h predicted results of Exp_Ctrl, Exp_SIC, Exp_SIC&SIT and observations.

| | Exp_Ctrl | Exp_SIC | Exp_SIC&SIT |
|---|---|---|---|
| BGEP_2011B | 2.23 | 0.59 | 0.28 |
| BGEP_2011D | 3.21 | 0.74 | 0.32 |
| IMB_2011K | 2.72 | 1.63 | 0.43 |
| IMB_2011L | 1.46 | 1.33 | 0.14 |

Compared with the observations of the BGEP_2011B (Fig. 16a), the fluctuation trend of SIT decreasing first and then rising with time can be better captured in Exp_SIC&SIT. In contrast, the initial SIT field of Exp_SIC has not been corrected by observed SIT, resulting in a large initial error. Although the prediction error of SIT is reduced under the assimilation of observed SIC, it is still quite different from the observation. Compared with the observations of BGEP_2011D (Fig. 16b), the SIT of Exp_SIC&SIT generally tends to be underestimated and cannot present the peak value of observed SIT, while its averaged absolute deviation compared to observed data (0.32 m) is much smaller than Exp_SIC (0.74 m).



Among four groups of observations, the averaged absolute deviations between the forecast results and the observation at the
IMB_2011K buoy are the largest, as shown in Table 2. As an example, we analyse the results on 21 September, in which
time the absolute deviation between Exp_SIC&SIT and the IMB_2011K buoy is biggest. In fact, according to the predicted
sea ice extent on that day, the IMB_2011K buoy locates at the sea ice marginal area of the Beaufort Sea. According to the
results of the frequency distribution of SIT by BRMT in September in Sect. 4.1 and the variation law of SIT by BRMT in the
Beaufort Sea at the end of September in Sect. 4.3, there is a situation of overestimating the amount of thin ice and
continuously maintaining the melting state. Nevertheless, the real situation could be that the buoy was located in the non-
marginal area with thicker ice, or the SIT in the marginal area of the Beaufort Sea was underestimated by BRMT.

The IMB_2011L buoy located in the Central Arctic has been in operation since 13 September 2011, and a total 19 d data
have been obtained until 1 October. It is worth noting that the absolute deviations of SIT in Exp_SIC&SIT at the
IMB_2011L buoy is the smallest during four groups of observations, with a value of 0.14 m. This result is consistent with
the conclusion in Sect. 4.3. In other words, BRMT can better capture the variation trend and range of SIT in the Central
Arctic, so the accuracy of predicted SIT in the Central Arctic is higher than that in other regions when it is applied to the
numerical forecast of SIT as an assimilation observation field.

**Table 3.** Average absolute deviations of sea ice thickness (m) between 168 h predicted results of Exp_Ctrl, Exp_SIC, Exp_SIC&SIT and
observations

|  | Exp_Ctrl | Exp_SIC | Exp_SIC&SIT |
|---|---|---|---|
| BGEP_2011B | 2.25 | 0.25 | 0.22 |
| BGEP_2011D | 3.29 | 0.69 | 0.27 |
| IMB_2011K | 3.02 | 1.61 | 0.39 |
| IMB_2011L | 1.52 | 1.39 | 0.12 |

For the three experiments (Table 3), the average absolute deviations between the 168 h forecast results and the observed
values are also compared. The results show that, similar to the 24 h forecast results, only assimilating the SIC in the
Exp_SIC can reduce the average absolute deviation of SIT by more than 50 % compared with the Exp_Ctrl, except for the
IMB_2011L observation. Moreover, in Exp_SIC&SIT, jointly assimilating SIC and reconstructed SIT can further reduce the
average absolute deviation of SIT compared with Exp_SIC, especially in IMB_2011K and IMB_2011L, which can be
decreased by more than 80 %. Therefore, the improvement of SIT prediction accuracy by joint assimilation of SIC and
reconstructed SIT will not weaken with the increase of forecast time, which also means its short-term prediction performance
is stable. However, it should be noted that in Exp_SIC and Exp_SIC&SIT, the average absolute deviations of 168 h forecast
results are slightly smaller than those of 24 h forecast results (Tables 2 and 3). This may be attributed to the fact that only
SIC or SIT are assimilated in the initial field, while other variables such as sea ice velocity and sea surface temperature do

 

not match in dynamics. With the increase of forecast time, they are gradually adjusted and consistent through dynamic integration, resulting in the improvement of prediction accuracy.

### 5.2.3 Interaction between SIC and SIT

In order to deeply explore the role of BRMT based on the interaction between SIC and SIT in Exp_SIC&SIT prediction experiment, the BGEP_2011B mooring facility is taken as an example. A line segment AB is drawn in Fig. 17, which is the
550 km long. The position BGEP_2011B is taken as the midpoint of AB, which is 275 km away from A or B point. The position of point A is (80° 9.666′ N, 155° 20.322′ W, and the position of point B is (75° 51.126′ N, 144° 30.606′ W).

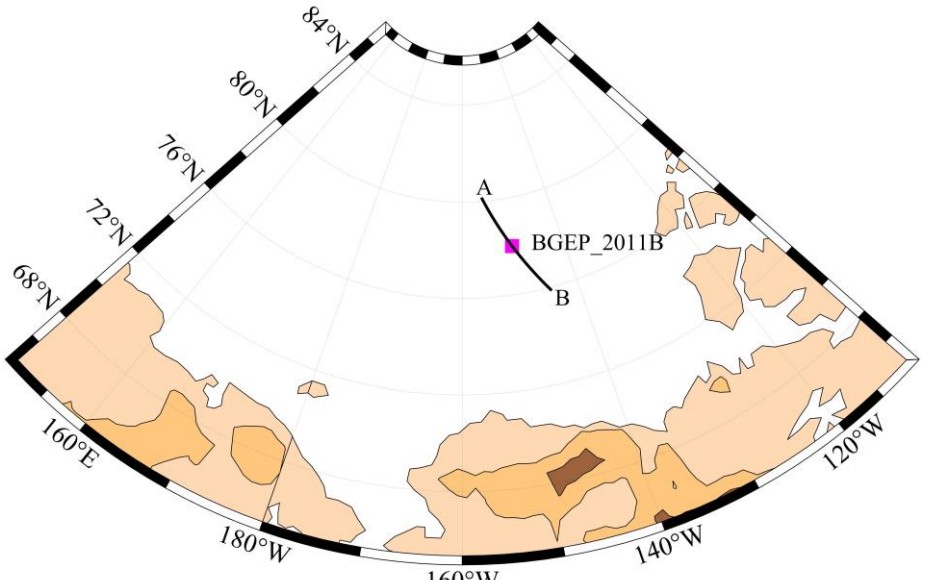

**Figure 17.** The schematic plot of mooring facility BGEP_2011B as the midpoint of line AB with a total length of 550 km.

In Fig. 18, the correlation coefficients along AB between the 24h forecast results of SIC and SIT in Exp_SIC and
Exp_SIC&SIT from 1-30 September 2011 is shown. The white grid indicates that there is no sea ice at this location. As can be seen from Fig. 18, both Exp_SIC and Exp_SIC&SIT show a positive correlation between SIC and SIT. It means that the larger the sea ice extent is, the thicker the SIT remains, which is consistent with the thermodynamics mechanism of sea ice proposed by Lisæter et al. (2003).

It is noted that in Fig. 18a, correlation coefficients between SIC and SIT within the range of 0-275 km are mostly above 0.75,
while there is a large extent of no sea ice within the range of 275-550 km and the rest values are mostly less than 0.75 except for the relatively high abnormal value of correlation coefficient due to the lack of data. This may be attributed that Exp_SIC only assimilates SIC, leading to poor dynamic coordination between SIC and SIT in the model integration process and the underestimation of sea ice extent. With the increase of the distance, the corresponding latitude gradually decreases and gets closer to the edge of the sea ice, so that the sea ice becomes thinner and its thermodynamic properties becomes more and





more localized, resulting in the weak correlation. This also explains why the 24 h predicted SIT of Exp_SIC is significantly

different from the BGEP_2011B and even the predicted SIT tends to be close to 0 after 12 September (Fig. 16a).





**Figure 18.** The correlation coefficients plot along AB between the 24 h forecast results of sea ice concentration and sea ice thickness in
Exp_SIC (a) and Exp_SIC&SIT (b) from 2 September to 1 October 2011 (Line AB intersects BGEP_2011B at 275 km (red heart)).

Similar to Fig. 18a, the correlation coefficients in the range of 0-275 km are higher than that in the range of 275-550 km in
Fig. 18b. However, the difference is that the correlation coefficient corresponding to 96.7 % grid points in Fig. 18b is 0.95 or
above, much higher than the 38.4 % in Fig. 18a. This result is not surprising. It is sufficient to indicate that the good
interactive relationship in BRMT between SIC and SIT has a positive impact on the coordination of the two in the model
integration process. In particular, it is noted that the predicted SIT of Exp_SIC&SIT almost coincide with the observations
from 7-12 September in Fig. 16a. Correspondingly, the correlation coefficient during this period between predicted SIC and
predicted SIT is close to 1 in the heart-shaped row of Fig. 18b. At the same time, the predicted SIT of Exp_SIC is much
larger than the observations (Fig. 16a) and the correlation coefficient of the heart-shaped line in Fig. 19a is obviously less
than that of Exp_SIC&SIT. This seems to give us a new inspiration. The correlation between the predicted SIC and the
predicted SIT is closely related to the quality of predicted SIT. This also explains the reason why BRMT based on the
interaction between SIC and SIT has a good performance in the forecast experiments of various sea ice elements.

## 6 Discussions and conclusions

In this study, a bivariate regression model is proposed to solve the problem that Arctic ice thickness in melting season cannot
be detected by satellite remote sensing technology. The regression model is established through using the reanalysis data of
SIC and SIT. Then, the SIT field can be constructed according to the SIC observational data at each grid point and the
corresponding regression model, named as BRMT.
From the distribution of averaged SIT during the multi-year melting seasons, the SIT from BRMT model is a better
compromise between other ice thickness data sets (CMST, PIOMAS and GIOMAS), which can describe the quantity and
hierarchical distribution of thin ice more accurately and will not overestimate the SIT in the thick ice area like CMST. At the
BGEP mooring facilities, BRMT basically reproduces the change of SIT in different years and months. The regression
model based on a single grid point makes the SIT of BRMT more fluctuating in a short period than CMST and PIOMAS,
which also makes BRMT more accurate to capture the change trend in a period of time. In particular, the lower RMSE and
bias of BRMT at BGEP_A and BGEP_D positions show that the overall performance of the SIT from BRMT during the
melting seasons is slightly better than that of other data sets. In the comparison of the deviation between each data set and 25
IMB buoys located in different regions, the SIT from BRMT, which has multi-year historical correlation information of SIC
and SIT, has significantly better consistency with in situ observation for both MYI and FYI in the Central Arctic than CMST
and PIOMAS. In addition, the outstanding performance of the SIT from BRMT in the E Greenland Sea and in parts of the
Beaufort Sea cannot be ignored.
Although the SIT from BRMT has gratifying highlights compared with other SIT data sets, it still has some shortcomings,
such as the overestimation of SIT at BGEP_B mooring equipment, the lag response to the initial melting state and the



stagnation to maintain the melting state at the end of melting season. The discrepancy between the SIT from BRMT and in situ observed SIT may come from the deviation of the reanalysis data set selected by the model and the coarse grid resolution of the SIC data from satellite remote sensing observation. As a result, there are few grid points with SIT value near the in situ observation position, so that the true SIT at this position is replaced by inaccurate weighted SIT at other

positions within the influence radius, resulting in large errors. In the supplementary experiment, the AMSR-E satellite remote sensing observation SIC data with a resolution of 12.5 km (https://nsidc.org/data/ae_si12/versions/3) is used in the bivariate regression model of SIT to replace the observation with the original resolution of 25 km. Then, RMSEs between the newly constructed SIT from BRMT and 25 IMB buoys are simply compared with those of the original constructed SIT, CMST and PIOMAS. Results show that the newly constructed SIT are better than those from the other three data sets in

seven buoys (Table A3).

In addition to the above effects caused by data errors, in fact, the relationship between SIC and SIT is nonlinear and complicated. Generally, the same SIC value corresponds to multiple SIT values and the range of thickness values stays large, so it is a little difficult to describe the relationship between them only using the fitting relationship. The ability of neural network in self-learning and high-speed searching for optimal solutions is extremely suitable for building the nonlinear

model. However, whether the performance of DL can be improved depends on the size of the data set. The more parameters DL model will learn, the more data and computing costs required for training will also increase. Otherwise, problems with more dimensions and small data will lead to over fitting of the network. On the premise of high spatial resolution, it needs millions or even tens of millions of samples to train a usable network for Arctic sea ice (Chi and Kim, 2017; Andersson et al., 2021). In contrast, the bivariate regression model proposed in this study only needs about 1400 samples, which greatly saves

the calculation time and cost. In the future, we expect to further explore the advantages and disadvantages between DL and statistical methods in constructing SIT, and even consider more elements related to SIT and introduce them into DL model, so as to provide more reliable references for the record of Arctic SIT in the melting season.

Furthermore, the forecast experiments in Arctic sea ice melting season are carried out when only assimilating SIC and jointly assimilating both SIC and the SIT from BRMT. Results of two experiments indicate the bivariate assimilation scheme

(hereinafter referred to as Exp_SIC&SIT) shows good stability and accuracy in the short-term forecast of Arctic SIC and SIT. Especially, the marginal ice zone predicted by Exp_SIC&SIT is basically consistent with observation and it can accurately capture the fine structure of sea ice. In addition, in the Central Arctic, the averaged absolute deviation between the predicted SIT and the observed SIT is only 0.14 m at the IMB position, which is far less than 1.33 m in the Exp_SIC. This result is consistent with the conclusion that BRMT is able to capture the variation trend of SIT in the Central Arctic and better agree

with the observation. We also found that the correlation between the predicted SIC and predicted SIT is mostly 0.95 or above in Exp_SIC&SIT. These results strongly demonstrate that the BRMT has certain application prospects and can be widely used in the Arctic sea ice melting season.

Finally, there are interactions and constraints among the sea ice and ocean. For the problems of how they interact and coordinate with each other in the ice-sea coupled model, the further in-depth analysis will be carried out from the perspective



of dynamic and thermal processes of sea ice. It is hope to improve the constructed model of SIT during Arctic sea ice
melting season and the physical process of the numerical model, so as to realize the accurate forecast in the Arctic sea ice
and marine environment elements.

**Appendix**

**Table A1.** The weights of certain years from 2004 to 2018 relative to 2011, 2012, 2013, 2014 and 2015, respectively. Cal_year=The year
used to calculate the weight. Tar_year=Target year. Nan=None data.

| Tar_year Cal_year | 2011 | 2012 | 2013 | 2014 | 2015 |
|---|---|---|---|---|---|
| 2004 | 0.0156 | 0.0124 | 0.0106 | Nan | Nan |
| 2005 | 0.0224 | 0.0163 | 0.0135 | 0.0113 | Nan |
| 2006 | 0.0338 | 0.0229 | 0.0180 | 0.0145 | 0.0122 |
| 2007 | 0.0518 | 0.0342 | 0.0248 | 0.0194 | 0.0158 |
| 2008 | 0.0793 | 0.0521 | 0.0357 | 0.0267 | 0.0213 |
| 2009 | 0.1200 | 0.0795 | 0.0532 | 0.0376 | 0.0295 |
| 2010 | 0.1771 | 0.1200 | 0.0803 | 0.0546 | 0.0414 |
| 2011 | Nan | 0.1769 | 0.1204 | 0.0821 | 0.0590 |
| 2012 | 0.1771 | Nan | 0.1769 | 0.1241 | 0.0862 |
| 2013 | 0.1200 | 0.1769 | Nan | 0.1844 | 0.1299 |
| 2014 | 0.0793 | 0.1200 | 0.1769 | Nan | 0.1943 |
| 2015 | 0.0518 | 0.0795 | 0.1204 | 0.1844 | Nan |
| 2016 | 0.0338 | 0.0521 | 0.0803 | 0.1241 | 0.1943 |
| 2017 | 0.0224 | 0.0342 | 0.0532 | 0.0821 | 0.1299 |
| 2018 | 0.0156 | 0.0229 | 0.0357 | 0.0546 | 0.0862 |

**Table A2.** RMSEs (m) of sea ice thickness among BRMT, CMST, PIOMAS and in situ sea ice thickness observations of 25 IMB buoys
(The date format is yyyy/mm/dd). MYI=Multi-year ice; FYI=First-year ice; CA=Central Arctic; B=Beaufort Sea; C=Chukchi Sea;
L=Laptev Sea; E=E Greenland Sea. Nan=None data. The underline indicates that the RMSE is the smallest of the three datasets.

| IMB buoy | Data range | Ice type | Region | BRMT | CMST | PIOMAS |
|---|---|---|---|---|---|---|
| 2011C | 2011.7.1-8.11 | MYI | CA | 0.2473 | 0.9713 | 0.3327 |
| 2011I | 2011.8.5-8.23 | MYI | B | 0.2204 | 0.4265 | 0.2170 |
| 2011J | 2011.8.6-9.30 | MYI | B | 1.4817 | 1.4031 | 1.1522 |
| 2011K | 2011.8.9-8.14/9.1-9.30 | MYI | B | 0.3918 | 0.2727 | 0.1525 |
| 2011L | 2011.9.13-9.30 | MYI | CA | 0.1685 | 0.3353 | 0.1181 |





| IMB buoy | Data range | Ice type | Region | BRMT | CMST | PIOMAS |
|---|---|---|---|---|---|---|
| 2012D | 2012.7.11-7.25/8.6-9.30 | FYI | CA | 0.7792 | 1.7555 | 1.6248 |
| 2012G | 2013.7.1-8.11/8.14-8.15/8.18-9.6 | FYI | CA | 0.4186 | 0.4595 | 0.3503 |
| 2012H | 2012.9.10-30/2013.7.1-9.26 | FYI | B | 0.3202 | 0.2897 | 0.5538 |
| 2012I | 2012.8.14-9.30 | MYI | C | 0.1599 | 0.4651 | 0.1813 |
| 2012J | 2012.8.25-9.30/2013.7.1-8.3 | MYI | L | 0.5367 | 0.4105 | 0.2808 |
| 2012L | 2012.8.27-9.30/2013.7.1-8.28 | MYI | B | 1.2109 | 1.4096 | 1.7758 |
| 2012M | 2012.8.29-9.29/2013.7.1-8.14 | MYI | E | 1.2163 | 1.5608 | 1.6098 |
| 2013B | 2013.7.1-9.30 | MYI | CA | 0.2580 | 0.5858 | 0.4596 |
| 2013F | 2013.8.25-9.30/2014.7.1-9.30 | MYI | B | 0.4353 | 0.2780 | 0.2082 |
| 2013G | 2013.9.4-9.30 | MYI | B | 0.6005 | 0.7842 | 1.2323 |
| 2013H | 2013.9.3-9.30 | MYI | CA | 0.3147 | 0.2868 | 0.4797 |
| 2014B | 2014.7.1-7.29 | FYI | B | 0.4886 | 0.4134 | 0.3804 |
| 2014C | 2014.7.1-8.24 | FYI | B | 0.2169 | 0.2181 | 0.6470 |
| 2014D | 2014.7.1-7.14/7.16-7.31 | MYI | CA | Nan | 0.6852 | 1.1991 |
| 2014F | 2014.9.17-9.30 | MYI | B | 0.2649 | 0.3026 | 0.5098 |
| 2014I | 2015.7.1-8.22 | MYI | B | 0.3314 | 0.1790 | 0.2441 |
| 2015D | 2015.7.1-8.29/9.1-9.30 | MYI | CA | 0.6678 | 1.2153 | 0.9346 |
| 2015E | 2015.7.1-7.6 | FYI | B | 0.2461 | 0.9598 | 0.6130 |
| 2015F | 2015.8.13-9.30 | MYI | CA | 0.2493 | 0.2306 | 0.4785 |
| 2015G | 2015.9.13-9.30 | MYI | CA | 0.2014 | 0.1169 | 0.3509 |

**Table A3.** RMSEs (m) of sea ice thickness among N-SIT, BRMT, CMST, PIOMAS and in situ sea ice thickness observations of 25 IMB buoys (The date format is yyyy/mm/dd). N-SIT=Newly con-structed sea ice thickness, BRMT=original constructed sea ice thickness. Nan=None data. The underline indicates that the RMSE is the smallest of the four datasets.

| IMB buoy | N-SIT | BRMT | CMST | PIOMAS |
|---|---|---|---|---|
| 2011C | 0.1931 | 0.2473 | 0.9713 | 0.3327 |
| 2011I | 0.3023 | 0.2204 | 0.4265 | 0.2170 |
| 2011J | 0.9473 | 1.4817 | 1.4031 | 1.1522 |
| 2011K | 0.5245 | 0.3918 | 0.2727 | 0.1525 |
| 2011L | 0.1534 | 0.1685 | 0.3353 | 0.1181 |
| 2012D | 1.0438 | 0.7792 | 1.7555 | 1.6248 |
| 2012G | 0.6090 | 0.4186 | 0.4595 | 0.3503 |
| 2012H | 0.7053 | 0.3202 | 0.2897 | 0.5538 |





| IMB buoy | N-SIT | BRMT | CMST | PIOMAS |
|---|---|---|---|---|
| 2012I | 0.4515 | 0.1599 | 0.4651 | 0.1813 |
| 2012J | 0.6155 | 0.5367 | 0.4105 | 0.2808 |
| 2012L | 0.9049 | 1.2109 | 1.4096 | 1.7758 |
| 2012M | 0.6157 | 1.2163 | 1.5608 | 1.6098 |
| 2013B | 0.2740 | 0.2580 | 0.5858 | 0.4596 |
| 2013F | 0.6592 | 0.4353 | 0.2780 | 0.2082 |
| 2013G | 0.3690 | 0.6005 | 0.7842 | 1.2323 |
| 2013H | 0.2186 | 0.3147 | 0.2868 | 0.4797 |
| 2014B | 0.9087 | 0.4886 | 0.4134 | 0.3804 |
| 2014C | 0.6073 | 0.2169 | 0.2181 | 0.6470 |
| 2014D | Nan | Nan | 0.6852 | 1.1991 |
| 2014F | 0.1265 | 0.2649 | 0.3026 | 0.5098 |
| 2014I | 1.0065 | 0.3314 | 0.1790 | 0.2441 |
| 2015D | 0.9383 | 0.6678 | 1.2153 | 0.9346 |
| 2015E | 1.0604 | 0.2461 | 0.9598 | 0.6130 |
| 2015F | 0.3197 | 0.2493 | 0.2306 | 0.4785 |
| 2015G | 0.2140 | 0.2014 | 0.1169 | 0.3509 |

**Data availability**

The SIC observational data set is available at the NSIDC (https://nsidc.org/data/NSIDC-0051/versions/1and https://nsidc.org/data/AU_SI12/versions/1). The sea ice draft data from the ULS measurements of BGEP are available at the WHOI
(https://www2.whoi.edu/site/beaufortgyre/data/mooring-data/2011-2012-mooring-data-from-the-bgep-project/), and the IMB data from are available from the CRREL-Dartmouth Mass Balance Buoy Program (http://imb-crrel-dartmouth.org/archived-data/). The SST data are available at ESA Climate Change Initiative's Sea Surface Temperature (https://climate.esa.int/en/odp/#/project/sea-surface-temperature). The SIT grid data from PIOMAS and GIOMAS are available at PSC (http://psc.apl.uw.edu/research/projects/arctic-sea-ice-volume-anomaly/data/model_grid and http://psc.apl.uw.edu/data/global-sea-ice-giomas-
data-sets/). The Arctic combined model and satellite SIT dataset is available at PANGAEA (https://doi.pangaea.de/10.1594/PANGAEA.891475). The reanalysis data of SIC and SIT are available at CMEMS(https://resources.marine.copernicus.eu/?option=com_csw&task=results?option=com_csw&view=details&product_id=ARCTIC_REANALYSIS_PHYS_002_003).



**Author contribution**

LY and XZ designed the method and experiment, LY and HF prepared the manuscript, XL and SZ helped interpret the
analysis and comment on this manuscript.

**Competing interests**

The authors declare that they have no conflict of interests.

**Acknowledgements**

The authors would like to thank the following data providers: the NSIDC for providing SIC data (https://nsidc.org/), the
Woods Hole Oceanographic Institution (WHOI) for providing sea ice draft data (https://www.whoi.edu/), the Cold Regions
Research and Engineering Laboratory for IMB data (http://imb-crrel-dartmouth.org/), European Space Agency (ESA)
Climate Change Initiative for providing SST data (https://climate.esa.int/en/), Copernicus Marine Environment Monitoring
Service (CMEMS) for providing the reanalysis data (http://marine.copernicus.eu), Polar Science Center (PSC) for providing
the SIT gird data (http://psc.apl.uw.edu/), Longjiang Mu from Sun Yat-sen University for providing the CMST SIT. This
study is funded by the National Key R&D Program of China (2018YFC1407401) and the Open Fund Project of Key
Laboratory of Marine Environmental Information Technology, Ministry of Natural Resources of the People's Republic of
China.

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
