# Peer review of "Reconstruction of Arctic sea ice thickness and its impact on sea ice forecasting in the melting season"

_The Cryosphere, 2022_

## Referee Comment (RC2)

[revised manuscript text omitted]

Compared with the observations of the BGEP_2011B (Fig. 16a), the fluctuation trend of SIT decreasing first and then rising with time can be better captured in Exp_SIC&SIT. In contrast, the initial SIT field of Exp_SIC has not been corrected by observed SIT, resulting in a large initial error. Although the prediction error of SIT is reduced under the assimilation of observed SIC, it is still quite different from the observation. Compared with the observations of BGEP_2011D (Fig. 16b), the SIT of Exp_SIC&SIT generally tends to be underestimated and cannot present the peak value of observed SIT, while its averaged absolute deviation compared to observed data (0.32 m) is much smaller than Exp_SIC (0.74 m).

[revised manuscript text omitted]

---

## Author Comment (AC1)

Author's point-to-point response on Referee Comment #1 to tc-2022-92. The reviewer comments appear in black. The responses are in blue and the proposed changes to manuscript are in *Arial*. L# refers to that in the track-changes file.

Introduction:

The current study presents a statistical method that uses SIC and SIT from a reanalysis dataset to construct a historical SIT dataset. The idea behind this is that the SIT of the source reanalysis dataset is not accurate in the melt season as no SIT measurements in the melt season are available to feed into that reanalysis dataset, and that incorporating statistical relationships between SIC and SIT leads to an improved SIT dataset. Detailed comparisons with in situ observations and other often-used SIT datasets show that the newly developed SIT performs well. In addition, assimilation runs are performed in which only SIC or both SIC and the newly constructed SIT dataset are assimilated, which are then used to initialize 7-day forecasts. The skill of forecasts initialized from the assimilation runs in which both SIC and the new SIT dataset are assimilated is shown to be higher than forecasts initialized from assimilation runs in which only SIC was assimilated. The analysis shown is detailed and interesting, but there are several major issues that the authors have to address before I can recommend publication.

**Major comments:**

1.  Although I understand that the authors are not native English speakers, the English is poor, which makes it difficult to follow the text. I suggest the authors improve the language by consulting with an English native speaker.

Thanks for your advice. We carefully modified the grammar of the sentence and the content of the manuscript in detail.

2.  Perhaps it is due to my lack of expertise in the area, or the poor English (or a combination of these factors), but I do not fully understand the statistical model that is used to construct the SIT as described in section 3.1 and in figure 2. Other readers may

have similar problems and therefore this should be improved. In particular, I do not understand the 'linear regression for each grid point'. What is particularly confusing is that the authors write that (l. 145) 'the linear regression process is carried out at each grid point … for each year.' And (l. 147) 'the corresponding SIC-SIT regression ... can be obtained for each year'. This description suggest that the linear regression is done spatially for each year (i.e. regression of SIT at a location with SIC at all other locations in a fixed year), but other text later in the paper suggests the linear regression is done at a specific grid point over the time dimension (i.e. regression of SIT at a location with SIC at that same location over time). Figure 2 also suggests that the regression is done spatially for each year, but I don't think that is what the authors mean. Please clarify.

Thank you for pointing this out. The linear regression we proposed here is done at a specific grid point over the time dimension, and is not done spatially for each year. In fact, what L145 and L147 try to explain is that for the proposed sea ice thickness field on July 1, 2011, we need to carry out linear regression for each non null value point on the reanalysis grid on July 1 of each year (from 2004 to 2018, except 2011). In other words, the independent variable is the reanalysis sea ice concentration value of a grid point on July 1 of a certain year, and the dependent variable is the reanalysis sea ice thickness value of the corresponding grid point. In this way, the SIC-SIT regression relationship at each grid point can be obtained for each year. We modified it in the revised manuscript. In addition, what we want to express in Figure 2 is also the regression in the time dimension, not the spatial dimension. We modified the second step and the third step in Figure 2 in the revised manuscript.

*"Then, starting from 2004, the linear regression process is carried out at each grid point (non null point) of the reanalysis data grid, with SIC of a grid point being as the independent variable and SIT of the grid point corresponding to SIC being as the dependent variable. The same linear regression process is performed in other years. The corresponding SIC-SIT regression relation at each grid point can be obtained for each year."*

[Figure]

**Revised Figure 2.** A flow chart of bivariate regression model of SIT (using 1 July 2011 as an example).

3. The abstract should be improved, as I initially did not understand the method that the authors are introducing. I understood that the aim of the authors is to construct a historical SIT dataset, but they aim to do that based on gridded SIC and It should be explained more clearly how the SIT that is the input of the BRMT method differs from the SIT output. Also, the abstract contains several statements about improved performance, without specifying the baseline:

- 17: 'BRMT-constructed SIT is more accurate': more accurate than what?

- 19: 'closer to observations': closer than what?

- 21: 'significantly improved': compared to what?

The baseline should be specified. Finally, some more details on the forecasting experiments should be included in the abstract. In particular, it would be helpful to note that these pertain to 7-day forecasts (to contrast with seasonal forecasts that run for up to a year and in the context of which SIT initialization is often discussed).

Thanks for your suggestion.

First of all, SIC and SIT of TOPAZ4 reanalysis data are used to construct the statistical relationship between SIC and SIT in BRMT method. Then, satellite observed SIC data is used as input and BRMT-constructed SIT is the output. I'm not sure whether "the SIT that is the input of the BRMT method" you mentioned refers to the SIT of TOPAZ4. In fact, we compared the SIT of BRMT, the SIT of TOPAZ4 and the site SIT in Figure 5 and Figure 6 in Section 4.2. The results show that the SIT of BRMT is closer to the observation than that of TOPAZ4. In addition, according to your suggestion in the Minor comments 2, we added the forecast experiment results with both the SIT of TOPAZ4 and the SIC from satellite remote sensing as the initial fields in Section 6. We hope to show the difference between the SIT of TOPAZ4 and the SIT of BRMT in a more comprehensive way. The conclusion of this part is supplemented in the abstract. Secondly, we modified the several statements about improved performance without specifying a baseline in the abstract.

Finally, we supplemented the conclusion of the forecast experiments in the abstract.

**Minor comments** (note: there are many more grammatical errors that I don't list below, see main comment #1):

1.   Some key references are omitted and should be included: Dirkson et al 2015 (https://doi.org/10.1002/2015GL063930.1) develops 3 statistical methods to generate a SIT datasets, and Dirkson et al 2017 (https://doi.org/10.1175/JCLI-D-16-0437.1) shows that one of the 3 statistical methods leads to improved seasonal forecasts of SIC. The authors should include these references.

Thanks. We included quotations from these two papers in the introduction of Section 1.

*"Dirkson et al. (2015) employed maximum covariance analysis to identify patterns of covariability between SIT and two predictors (SIC and lagged sea level pressure) by considering the thermodynamic and dynamic relationship among them, so as to realize the real-time estimation of SIT. On this basis, the prediction performance of sea ice area and regional SIC to some extent was improved by the SIT initialization field generated based on the further improved statistical model through extrapolation (Dirkson et al., 2017)."*

2.   The authors try to highlight the importance of the newly SIT dataset by comparing forecasts initialized from assimilation run in which it is used with forecasts initialized from assimilation runs in which it is not used. While this is interesting and worth reporting, it only highlights the importance of initializing SIT versus not initializing SIT. To investigate whether or not the newly developed SIT dataset provides additional value compared to other SIT datasets (e.g. that from the reanalysis dataset that it was derived from) in the context of forecasts, an additional set of forecasts would have to be presented in which an alternative SIT dataset is used for creating the initial conditions.

Thanks for your suggestion. The new forecast experiment in September 2011 with both the SIT of TOPAZ4 and the SIC from satellite remote sensing as initial fields is supplemented (named Exp_TOPAZ). Comparing the forecast results of the two

experiments, it can be seen that Exp_SIC&SIT is significantly closer to the observations than Exp_TOPAZ in terms of the RMSE between SIC forecast results and observations in the whole forecast period (Fig. 1), or in terms of 1d or 7d forecast results of sea ice extent (Fig. 2). In the comparison between the 1 d SIT forecast results of the two experiments and the four situ observation facilities (Table 1), the average absolute deviations between Exp_SIC&SIT and two observation facilities (BGEP_2011D and IMB_2011L) are 39% and 67% lower than that of Exp_TOPAZ, while the average absolute deviations between Exp_SIC&SIT and the other two observation facilities (BGEP_2011B and IMB_2011K) are 86% and 65% higher than that of Exp_TOPAZ, respectively. In general, the initial conditions provided by SIT from BRMT are expected to provide valuable reference results for Arctic sea ice forecast. The conclusion of this part is added to the discussion in Section 6.

[Figure]

**Figure 1.** Differences between the RMSEs of sea ice concentration forecast results of Exp_TOPAZ and Exp_SIC&SIT relative to the SSMI observation during the period of 2 September to 7 October 2011 (each segment represents the 7 d forecast). Date format is dd/mm.

[Figure]

**Figure 2.** Comparison of sea ice extent (unit: $10^6$ km$^2$) between 1 d (a) and 7 d (b) forecast results of Exp_SIC&SIT (red dotted line) and Exp_TOPAZ (light green dotted line) and the SSMI observations (solid blue line) in 2011, respectively. Date format is dd/mm.

**Table 1.** Average absolute deviations of sea ice thickness (m) between the 1d or 7d forecast results of Exp_SIC&SIT and Exp_TOPAZ and situ observations.

|  | 1d | | 7d | |
|---|---|---|---|---|
|  | Exp_SIC&SIT | Exp_TOPAZ | Exp_SIC&SIT | Exp_TOPAZ |
| BGEP_2011B | 0.28 | 0.15 | 0.22 | 0.13 |
| BGEP_2011D | 0.32 | 0.53 | 0.27 | 0.43 |
| IMB_2011K | 0.43 | 0.26 | 0.39 | 0.24 |
| IMB_2011L | 0.14 | 0.43 | 0.12 | 0.35 |

3. 63,66: is à was

Agreed. We corrected this.

4. 255: will not à does not?

Agreed. We corrected this.

5.   Figure 5 (bottom map): what do the colors represent?

The purpose of different mooring facilities marked by different colors is to make it easier to distinguish them in the figure. The colors have no specific meaning.

6.   Figure 8: I suggest to use a non-linear scale for the Normalized standard deviation as a) the most interesting data is where normalized standard deviation is close to 1, and because most points are located there

Thanks for your suggestion. In order to better explain "In the part with both NSTD and NCRMSD less than 5.5" in Section 4.3, we added Figure S1 in the supplementary document. At the same time, this figure can more clearly show the dense data points in Figures 8a-8c, and also make the points with the normalized standard deviation close to 1 more obvious.

[Figure]

**Figure S1.** Taylor diagrams (the part of both NSTD and NCRMSD less than 5.5) of (a) BRMT, (b) CMST and (c) PIOMAS with respect to all available IMB buoy data during melting seasons from

2011 to 2015. The green dotted lines indicate the normalized CRMSD. The reference observations are indicated by Ref in red.

7. 365: 'In part with small deviation evaluation criteria value': not clear. In the following lines (including the quoted numbers for correlation coefficient), do you only use data points with 'small' standard deviation, and if yes, what is the cut-off value for 'small'?

Thank you for pointing this out. The small deviation evaluation criteria value means that both NSTD and NCRMSD are less than 5.5. In the following lines, we used the points with both NSTD and NCRMSD less than 5.5 to calculate the evaluation criteria. We changed *'In part with small deviation evaluation criteria value'* to *'In the part with both NSTD and NCRMSD less than 5.5 (Figure S1)'* in the revised manuscript.

8. 477 'significantly smaller': this is a bit hard to see from Fig. 13 as it is hard to compare panel a with panel b. Perhaps it would make sense to add a 3rd panel showing the difference between panel a and b? Also: it is not clear what the authors mean with 'a long-term stable effect' in l. 479

Thank you for pointing this out. We delated the Figure 13a and Figure 13b and added the figure (Figure 13 in the revised manuscript) showing the differences between the RMSEs of 1 d and 7 d sea ice concentration forecast results of Exp_SIC and Exp_SIC&SIT relative to the SSMI observation during the period of 2 September to 7 October in 2011-2013. It can be calculated from the Figure 13 that the average difference between the RMSEs of the 1 d forecast results of Exp_SIC and Exp_SIC&SIT is about 0.0043 in 2011, and the average difference in 2011 of 7 d is about 0.0038. These two values are about 0.0025 and 0.0029 in 2012, 0.0026 and 0.0028 in 2013 respectively. These indicate that the improvement of the SIT initial field not only plays a significant role in improving the forecast accuracy of SIC, but also this improvement will not weaken with the increase of forecast time. We changed the description in the revised manuscript.

9.  Figure 14: it is hard to see what the authors refer to, as all the figures are so similar. Perhaps adding a contour line would help, but as it stands the current figure 14 does not add much to the paper. Figure 15 is much more informative

Thanks for your suggestion. Considering the low accuracy of SSMIS satellites in areas of low concentration or thin ice and the possibility that SSMIS may not be able to solve the problem of margin ice zones (MIZ), the MIZ analysis is deleted in the Section 5.2.1. In order to supplement the results of SIC, we added a new section to discuss the prediction results of sea ice extent (see Section 5.2.2 in the revised manuscript).

10. 530 'are largest': except for Exp_Ctrl

Thank you for pointing this out. We changed this in the revised manuscript.

11. 533: 'Variation law': not sure what is meant with that

Thank you for pointing this out. We didn't describe it clearly. We think this should be "Variation trend". It refers to that, according to Figure 10, the sea ice thickness of BRMT maintains the melting state at the end of September in the Beaufort Sea. We changed this in the revised manuscript.

Dirkson, A., Merryfield, W. J., Monahan, A: Real-time estimation of Arctic sea ice thickness through maximum covariance analysis, Geophys. Res. Lett., 42, 4869-4877, https://doi.org/10.1002/2015GL063930, 2015.

Dirkson, A., Merryfield, W. J., Monahan, A: Impacts of Sea Ice Thickness Initialization on Seasonal Arctic Sea Ice Predictions, J. Climate., 30, 1001-1017, https://doi.org/10.1175/JCLI-D-16-0437.1, 2017.

---

## Author Comment (AC2)

Author's point-to-point response on Referee Comment #2 to tc-2022-92. The reviewer comments appear in black. The responses are in blue and the proposed changes to manuscript are in *Arial*. L# refers to that in the track-changes file.

In this manuscript the authors present a statistically reconstructed dataset of Arctic summer sea ice thickness (SIT), which they create using SIT & SIC from the CMEMS Arctic reanalysis system, TOPAZ. The resulting SIT reconstruction, BRMT, is then compared with several model products, and thickness estimated by BGEP Eulerian moorings and IMB Lagrangian buoys. Finally, they BRMT dataset is assimilated into a short forecasting experiment for the period September 2011.

There are some interesting ideas and concepts here that are worthy of publication in The Cryosphere. However, there are some major issues that will need to be addressed before the manuscript can be published.

Thank you for your approval. We have carefully revised the issues in the manuscript in detail.

**Major comments/concerns:**

1.  The quality of English language used throughout the manuscript leaves a lot to be desired. As well as reducing the overall readability of the paper, there are also several cases in which it is hard to understand what is being said (or why).

I'm very sorry for the bad reading experience. We made detailed modifications to the manuscript.

2.  One of the main motivations listed for this study, and the BRMT dataset, is that summer satellite SIT data is not available. This issue is mentioned several times and the authors go so far as to state that it is "impossible". However, the authors do not take into account the fact that summer satellite SIT has been being developed now for many years. The first dataset of summer Arctic radar freeboard (10-years) was published at

the beginning of this year (Dawson et al., 2022). (NB. conversion from radar freeboard to SIT has also been done but that paper is still in press.)

I'm not necessarily saying that the existence of these new datasets invalidates the motivation for this study, but it should definitely be referenced and included in the discussion. How does/would the BRMT compare with the Dawson et al. (2022) freeboard?

Thanks for your suggestion. We ignored the latest research progress. All inappropriate expressions have been corrected in the revised manuscript. We tried to find the summer Arctic radar freeboard dataset online, but we couldn't find it. On July 28, 2022, we contacted Dawson and asked whether the dataset had been publicly accessed. Professor Dawson replied that the Arctic summer sea ice freeboard data is not currently accessible online, but they are going to release a more comprehensive dataset with Arctic summer freeboard, thickness data etc. So, in the discussion in Chapter 6, we cited the research of Dawson et al. (2022) and pointed out that when the new summer sea ice thickness (SIT) data set is released, it will provide a more reliable reference for evaluating the quality of BRMT.

[Figure]

**Figure 1.** The email reply of Dawson on whether the Arctic summer sea ice freeboard data is accessible online.

In addition, although the radar freeboard data are not available at present, in Figure 5 of Dawson et al. (2022), the author gives the diagrams of Cryosat-2 summer (May-September) radar freeboards for 2013 (hereinafter referred to as the reference figure). Since the longitude and latitude information of the reference figure cannot be obtained, we try to roughly plot the SIT of BRMT from July to September 2013 according to the geographical location of the reference figure. Here, according to the time period division of the reference figure, July to September is divided into six time periods, which are July 1st-15th, July 16th-31st, August 1st-15th, August 16th-31st, September 1st-15th, and September 16th-30th. The six Arctic SIT figures were processed into the same pixel size as the reference figures (Figure 2).

[Figure]

**Figure 2.** Arctic radar freeboard and sea ice thickness from July to September 2013, including six time periods: (a) July 1st-15th, (b) July 16th-31st, (c) August 1st-15th, (d) August 16th-31st, (e) September 1st-15th,

(f) September 16th-30th. In each group of figures, the left figure represents the radar freeboard from Dawson et al. (2022), and the right figure represents the sea ice thickness of BRMT.

It can be seen from Fig. 2 that the figures of both radar freeboard and SIT show that sea ice began to melt from the beginning of July until the sea ice extent reached the minimum in mid-September, and then gradually frozen. Although there is a strong correlation between radar freeboard and SIT, there are also some differences between them in the period of rapid change of sea ice melting and freezing. From mid-August to early September, the figures of radar freeboard show that although the sea ice extent was still shrinking, the sea ice gradually frozen from the north of the Canadian Arctic Archipelago, and the freezing extent extended to most areas of the Beaufort Sea by the end of September. However, the figures of SIT show that the sea ice had been melting from mid-August to early September. Until the mid-September, the sea ice in the Beaufort Sea showed obvious thickening, and the sea ice extent expanded accordingly. At present, we cannot obtain accurate summer radar freeboard or SIT data set covered Arctic, so it is difficult to properly compare between the radar freeboard from Cryosat-2 and the SIT from BRMT.

**Table 1.** The correlation coefficients, mean bias and standard deviation between CryoSat-2 / BRMT and ULS sea ice drafts for Moorings A, B and D, respectively.

|  | Correlation coefficients | The mean bias and standard deviation |
| --- | --- | --- |
| BGEP_A | 0.76 / 0.85 | $-0.13\pm0.45$m / $0.15\pm0.63$m |
| BGEP_B | 0.63 / 0.83 | $-0.33\pm0.52$m / $0.33\pm0.61$m |
| BGEP_D | 0.62 / 0.84 | $-0.29\pm0.51$m / $0.07\pm0.68$m |

In Dawson et al. (2022), the author converted Cryosat-2 radar freeboards to estimates of sea ice draft and compared it with Beaufort Gyre Exploration Programme Mooring ULS (Upward Looking Sonar). Table 1 shows the correlation coefficients, mean bias and standard deviation between CryoSat-2 (between May and September, 2011-2018), BRMT (between July and September, 2011-2015) and ULS sea ice drafts for Moorings

A, B and D, respectively. The results show that the standard deviation of Cryosat-2 is smaller than that of BRMT, but the correlation coefficient of BRMT is higher.

3. The paper manages to be both too long and detailed, and too short and vague at the same time. By this I mean that, despite the paper being rather long, sufficient details are not provided of the methods used to create BRMT. Given the length of the paper, I recommend that the authors consider dropping Section 5 and focusing properly on the BRMT reconstructed dataset – both creation and evaluation.

Thank you for your suggestion. The original intention of BRMT method is to provide sufficient SIT observation data to realize the correction of initial field in summer sea ice prediction. When carrying out real-time prediction, the reanalysis data set only has historical SIT data, which cannot meet the needs of real-time assimilation. However, the BRMT model only needs satellite observation of sea ice concentration (SIC) to estimate the SIT observation in real-time. Therefore, the forecast experiment in Section 5 is also to examine whether the forecast results of sea ice variables have improved when the assimilation of SIT from BRMT is introduced in addition to the assimilation of SIC. This provides strong evidence for the effectiveness of applying SIT from BRMT to real-time prediction. Considering the length of the paper, we only made a prediction for one month in a year. However, as you mentioned in Major comment 4, the forecast experiment results in more time periods are needed to fully illustrate the role of SIT from BRMT. Therefore, we have increased the period of forecast experiment. See the reply in Major comment 4 for the detailed results. At the same time, we have also modified the content of both creation and evaluation of BRMT. See the reply of Major comment 6 for the detailed results.

4. The "retrospective forecast experiments" in Section 5 are really only cursory, with less than one month of forecasts performed for only one particular year. To properly assess the impact of assimilating the BRMT SIT a much more comprehensive assessment would be needed – including the whole summer period on at least 2 different years. Many would also question the fact that BRMT uses information from the future

(i.e., 2012-2018) in the reconstruction. This makes it unusable for real-world forecasting situations. How much skill would be lost if you were to only use past data?

Thank you for your advice. Considering the refinement of the article and the fact that September is a period of rapid changes in Arctic sea ice melting and freezing, we still choose September for analysis in the "retrospective forecast experiments" in Section 5, and expand the time range to 2011-2013. In addition, according to the comment of Minor comment 3, we added the analysis of the prediction results of the sea ice extent. The adjusted Section 5 is divided into the forecast results of SIC, sea ice extent and SIT, respectively. The revised section is more logical and comprehensively evaluates the role of SIT from BRMT.

Thank you for your suggestions on the construction process of BRMT. In fact, Figure 2 in the manuscript just wants to show the algorithm flow of BRMT model. The specific selection of years in reanalysis data should be determined according to the target year of construction. But we did ignore that BRMT uses information from the future in the reconstruction. Therefore, we reconstructed the Arctic SIT from July to September in 2011-2015 based on BRMT model without using future information (excluding the future years in table A1 in the manuscript), named BRMTnew. The reanalysis data selected for different target years and the corresponding weights are shown in Table 2.

**Table 2.** The weights of certain years from 2004 to 2014 relative to 2011, 2012, 2013, 2014 and 2015, respectively. Cal_year=The year used to calculate the weight. Tar_year=Target year. Nan=None data.

| Tar_year / Cal_year | 2011 | 2012 | 2013 | 2014 | 2015 |
|---|---|---|---|---|---|
| 2004 | 0.0312 | 0.0236 | 0.0183 | Nan | Nan |
| 2005 | 0.0448 | 0.0327 | 0.0247 | 0.0183 | Nan |
| 2006 | 0.0676 | 0.0477 | 0.0350 | 0.0247 | 0.0183 |
| 2007 | 0.1036 | 0.0709 | 0.0507 | 0.0350 | 0.0247 |
| 2008 | 0.1587 | 0.1059 | 0.0739 | 0.0507 | 0.0350 |
| 2009 | 0.2399 | 0.1572 | 0.1075 | 0.0739 | 0.0507 |
| 2010 | 0.3543 | 0.2307 | 0.1555 | 0.1075 | 0.0739 |
| 2011 | Nan | 0.3313 | 0.2223 | 0.1555 | 0.1075 |
| 2012 | Nan | Nan | 0.3123 | 0.2223 | 0.1555 |
| 2013 | Nan | Nan | Nan | 0.3123 | 0.2223 |
| 2014 | Nan | Nan | Nan | Nan | 0.3123 |

Firstly, the qualitative comparison of error for the SIT from BRMT and BRMTnew relative to BGEP mooring facilities (BGEP_A, BGEP_B, BGEP_D) is quantified. The details of RMSE and Bias in 2011-2015 are shown in Fig. 3. The results show that except for 2013A, 2013B, 2014A and 2015A, the differences of RMSE and bias between BRMT and BRMTnew are less than 0.05 m. In general, the average RMSE of BRMTnew is 0.0325 m more than that of BRMT, and the average Bias of BRMTnew is 0.0316 m larger than that of BRMT.

[Figure]

**Figure 3.** RMSE versus bias during melting season from 2011 to 2015 (marked in different colors) for BRMT (△) and BRMTnew (○) relative to BGEP moorings BGEP_A (a), BGEP_B (b) and BGEP_D (c).

Then, we calculated the correlation coefficient and normalized STD and CRMSD of the SIT of BRMT and BRMTnew relative to the IMB buoy data according to the Taylor chart index mentioned in Section 4.3 of the manuscript. In general, most of the points of BRMT and BRMTnew are concentrated in the range of NSTD and NCRMSD less than 6. We only show the results of this part here (Figure 4). The results show that the results of the two indicators are very close. The average values of NSTD, NCRMSD and correlation coefficient of BRMTnew are 1.797, 1.419 and 0.685 respectively, and BRMT is 1.802, 1.423 and 0.692 respectively. The conclusions are supplemented in Section 6.

[Figure]

**Figure 4.** Taylor diagrams of (a) BRMT and (b) BRMTnew with respect to part of IMB buoy data during melting seasons from 2011 to 2015. The green dotted lines indicate the normalized CRMSD. The reference observations are indicated in red.

5.   I struggle with the concept and motivation for the reconstructed SIT dataset. BRMT essentially uses the relationship between SIC and SIT in the TOPAZ reanalysis. Much of the motivation for using TOPAZ in this way is not included and so I am left with so many questions in my head: So why not just use TOPAZ? What extra is BRMT bringing to the table? Why do you trust the relationships in TOPAZ so much? How much difference would using a different reanalysis make to the SIT reconstruction? I'm also concerned that there is some horrible kind of circularity in the analysis here, whereby desired traits – such as the relationship between SIC & SIT – are included in the design of the system and then used as part of the evaluation.

Thank you very much. As mentioned in major comment 3, the original intention of BRMT is to apply it to real-time sea ice prediction. If TOPAZ is used in real-time forecast, it will be almost impossible to achieve. Because TOPAZ reanalysis data is only historical data, and it cannot provide real-time summer SIT. However, the BRMT model can estimate the summer SIT in real-time based on the SIC satellite remote sensing data and it can be expected to improve the initial field quality of sea ice prediction. In addition, the reason why TOPAZ reanalysis data is trusted to build the relationship between SIC and SIT is that TOPAZ reanalysis data is obtained by numerical model, and all variables are constrained by physical equations, so the relationship between SIC and SIT is in line with physical laws. If there is summer SIT observation data covering the Arctic in the future, we also hope to use the observation

data of SIC and SIT to build this relationship, which may make the SIT estimated by BRMT more accurate.

Of course, at present, other reanalysis data sets can also be considered in the BRMT model to build the relationship between SIC and SIT. In order to test the impact of using other reanalysis data sets, we selected the GLORYS12V1 data set (named GLOBAL_MULTIYEAR_PHY_001_030, https://resources.marine.copernicus.eu/product-detail/GLOBAL_MULTIYEAR_PHY_001_030/INFORMATION). This product is the CMEMS global ocean eddy-resolving (1/12° horizontal resolution, 50 vertical levels) reanalysis covering the altimetry (1993 onward). It is based largely on the current real-time global forecasting CMEMS system. The model component is the NEMO platform driven at surface by ECMWF ERA-Interim then ERA5 reanalyses for recent years. Observations are assimilated by means of a reduced-order Kalman filter. Along track altimeter data (sea level anomaly), satellite sea surface temperature, SIC and in situ temperature and salinity vertical profiles are jointly assimilated. Moreover, a 3D-VAR scheme provides a correction for the slowly-evolving large-scale biases in temperature and salinity.

Without using future information (as shown in Table 2), we reconstructed the Arctic SIT from July to September in 2011 to 2015 based on BRMT model and CMEMS reanalysis data, named BRMTcmems.

Firstly, the qualitative comparison of error for the SIT from BRMTnew (mentioned in Major comment 4) and BRMTcmems relative to BGEP mooring facilities (BGEP_A, BGEP_B, BGEP_D) is quantified. The details of RMSE and Bias in 2011-2015 are shown in Fig. 5. The results show that except for 2011A, 2011B, 2011D, 2013D and 2014D, the differences of RMSEs and biases between BRMTnew and BRMTcmems are less than 0.2 m. In general, the average RMSE of BRMTcmems is 0.1734 m more than that of BRMTnew, and the average Bias of BRMTcmems is 0.2092 m larger than that of BRMTnew. In fact, there are obvious differences between the basic data sets (TOPAZ and CMEMS) used by the two and the BGEP mooring facilities (Figure 6). The difference of average RMSE between TOPAZ and CMEMS is about 0.3027 m,

and the difference of average bias is about 0.4147 m.

[Figure]

**Figure 5.** RMSE versus bias during melting season from 2011 to 2015 (marked in different colors) for BRMTnew (△) and BRMTcmems (○) relative to BGEP moorings BGEP_A (a), BGEP_B (b) and BGEP_D (c).

[Figure]

**Figure 6.** RMSE versus bias during melting season from 2011 to 2015 (marked in different colors) for TOPAZ (△) and CMEMS (○) relative to BGEP moorings BGEP_A (a), BGEP_B (b) and BGEP_D (c).

Then, we calculated the correlation coefficient and normalized STD and CRMSD of the SIT of BRMTnew and BRMTcmems relative to the IMB buoy data according to the Taylor chart index mentioned in Section 4.3 of the manuscript. In general, most of the points of BRMTnew and BRMTcmems are concentrated in the range of NSTD and NCRMSD less than 5. We only show the results of this part in Fig. 7. The results show that the average NCRMSD of BRMTnew is smaller than that of BRMTcmems, which are 1.4197 and 1.5289, respectively. In addition, the average correlation coefficient of

BRMTnew is also higher than that of BRMTcmems, which are 0.6854 and 0.5188, respectively. However, the average NSTD of BRMTcmems is slightly smaller than that of BRMTnew, which is 1.7364 and 1.7971 respectively.

In conclusion, the selection of different reanalysis data sets has a certain impact on the reconstructed SIT. From the current results, TOPAZ is not a disappointed choice. The conclusions are supplemented in Section 6.

[Figure]

**Figure 6.** Taylor diagrams of (a) BRMTnew and (b) BRMTcmems with respect to part of IMB buoy data during melting seasons from 2011 to 2015. The green dotted lines indicate the normalized CRMSD. The reference observations are indicated in red.

In the forecast experiment, we design and verify the correlation between SIC and SIT forecast results, which is mainly based on the following considerations. If the quality of the sea ice initial field can be effectively improved by the SIT from BRMT, the forecast results between SIC and SIT of the forecast experiment jointly assimilating SIC and SIT (Exp_SIC&SIT) should be more accurate than that of the forecast experiment assimilating only SIC (Exp_SIC). Correspondingly, the correlation between SIC and SIT forecast results will be better in Exp_SIC&SIT. It is inappropriate to simply attribute this result to the use of the correlation between SIC and SIT in BRMT. However, it must be admitted that we only select a short distance in section 5.2.3 to test the interaction between SIC and SIT, which is not comprehensive and persuasive. Because whether there is a high or low correlation between SIC and SIT in a certain range is closely related to whether the SIC is greater than 1 or less than 1. In other words, if the SIC is less than 1, the correlation may be relatively strong. If the SIC is

greater than 1, the correlation may be relatively weak. Therefore, based on the suggestions given by the reviewer, this section will be deleted in the revised manuscript.

6.  The comparison of the model-based and reconstructed SIT observations with the in-situ observations is either not performed carefully enough or not described adequately. I am not convinced that these comparisons are being performed or interpreted correctly for the following reasons:

Thank you for your professional advice. In Section 4, we modified and supplemented the comparison of the model-based and reconstructed SIT with the in-situ observations in more details. In the revised manuscript, the description and discussion of this part are more rigorous than before.

1)  The authors do not specify anywhere how they define Sea Ice Thickness (SIT). Is it the "floe thickness" (i.e., the average thickness of all the sea ice floes present in the grid-cell) or the "grid-box-mean thickness" (i.e., sea ice volume per unit grid-cell area)? The former is certainly what your point/in-situ observations (BGEP/IMB) are measuring. However, the latter is the prognostic used in the sea ice continuum models formulation and so very likely what you are using from the model-based products (TOPAZ/PIOMAS/GIOMAS/etc.). Obviously for SIC=1 these definitions are the same but not for SIC<1. Another consideration is that if using the "grid-box-mean thickness" definition, SIT will likely be much more correlated with SIC than using the "flow thickness" definition.

Thanks for your advice. The SIT of BRMT is based on the relationship between SIC and SIT constructed by model product (TOPAZ4), so its definition should be "grid-box-mean thickness". The missing definition is supplemented in Section 3.1 of the revised manuscript. In addition, the in-situ SIT observation (BGEP and IMB) selected in the manuscript belongs to "floe thickness", which is different from the "grid-box-mean thickness" of BRMT (or PIOMAS/CMST/TOPAZ) when the SIC is less than 1. This difference in definition is also pointed out in Section 4.2 of the revised manuscript.

In addition, the compared analyses between SIT of each model and in-situ observation are also modified in Sections 4.2 and 4.3.

2) There is no discussion of how much one would expect agreement between the model and the in-situ observations. In particular the BGEP data are point observations (of ice draft converted to thickness) and are being compared with the modelled thickness in a large grid-cell. Even for the case SIC=1, when the SIT definition issue above is not present, a direct comparison is not obvious. The same is true for the IMBs which will only ever model a single floe. Furthermore there is no discussion of the sampling issues one would expect in the IMB dataset. The IMBs are of course Lagrangian in nature and are permanently attached to the same ice floe. So one would expect changes in thickness to relatively slow, given that they are purely driven by thermodynamics. Meanwhile the dynamical nature of the Eulerian model could have huge changes in thickness from one time-step to the next. Finally, there are known sampling biases in the IMB dataset that should be discussed. IMBs are normally deployed just before the freeze-up in ice that has survived the summer. Mid-thickness floes are normally chosen – avoiding thick floes for practical reasons and thin floes to limit the chance of losing expensive equipment too quickly.

Thank you for pointing out the matters needing attention when comparing with the two in-situ observations (BGEP and IMB). Not only does the comparison of point observations and large grid-cell values exist in the SIT, but also in the ocean variables (such as temperature, salinity, sea surface height, etc.). In this case, the idea of data assimilation is usually adopted. The observation operator is used to project the model grid points onto the observation points, so that the model grid points become observable. In fact, the objective analysis method of Eq. 10 in the manuscript is essentially a data assimilation method, which processes the grid-cell SIT of BRMT (or CMST/PIOMAS/ TOPAZ) into a value that can be compared with the SIT of observation point (BGEP/ IMB). However, the difference between the two is unavoidable. The possible reasons include the error of the observation equipment itself, the error caused by the

simplification of the physical process of the model or the projection of grid points, and so on. Before there is no more suitable dataset of Arctic summer SIT that can be used for comparison, it is a common method compared with the SIT of point observations (Mu et al., 2018; Liang et al., 2019). This part of the discussion is supplemented in Section 4.2 of the revised manuscript. In addition, the discussion of changes in SIT caused by sampling issues in the IMB dataset is supplemented in Section 4.3. Further, the sampling biases in the IMB dataset are also discussed in Section 4.3.

**Minor comments/concerns:**

1.  In many cases results seem somewhat overstated. In particular the performance of BRMT in the East Greenland region, described as 'outstanding', is based upon comparison with only 2 IMB buoys – although the Lagrangian trajectory will include several individual measurements, they will only be of 2 individual ice floes! (NB. the same is true for the forecast results, which are based on only a few forecasts performed in a single year, but this is already mentioned above.)

Thanks for your advice. We indeed use inappropriate adjectives in some cases and it is not clear and accurate in the expression of some results. In the revised manuscript, we revised these issues. The problem of forecast results is modified in the manuscript and the reply is shown in Major comment 4.

2.  Too little information is provided about the model/reanalysis products being used. In particular, what observations are assimilated and what surface forcing is being for the reanalyses. This applies for both the reanalyses datasets in Section 2 and the MITgcm model in Section 3.

Thank you for pointing this out. More information about the reanalysis data set in Section 2 and the MITgcm model in Section 3 is supplemented in the revised manuscript.

3.  RMSE of SIC is not a very good metric for sea ice forecasts because of the errors in the passive microwave satellite observations. This is particularly true in the summer when the SSMIS cannot distinguish surface melting/ponds from open water. The SSMIS accuracy is also lower in areas of low concentration or thin ice. These points are the motivation for people using sea ice "extent" to compare with satellites.
In particular, I would drop the MIZ analysis in 5.2.1/Figure 15 because the SSMIS satellite is likely not able to resolve that. If you redid this analysis using AMSRE2 observations (which is higher resolution and more able to resolve thin/low concentration ice) then the results could be quite different.

Thanks for your suggestion. Although RMSE is not the best criteria to evaluate the forecast results of summer SIC, it is a relatively intuitive result. In Yang et al. (2015a, 2015b), the RMSE is calculated in summer (June-August) to compare the forecast results of the model and the SSMIS observation. In addition, SIC is one of the most important variables of sea ice, so we still intend to retain this part of the analysis. However, considering the low accuracy of SSMIS satellites in areas of low concentration or thin ice and the possibility that SSMIS may not be able to solve the problem of margin ice zones (MIZ), according to the opinions by the reviewer, is added and the MIZ analysis is deleted in the Section 5.2.1. In order to supplement the results of SIC, we added a new section to discuss the prediction results of sea ice extent (see Section 5.2.2 in the revised manuscript).

4.   Some of the figures do not bring any useful information and so could either be removed or reformulated. For example, the data in Fig 13 can be understood easily from Fig 12b. Similarly, Fig 14, for which all the panels in look the same, could be improved by changing the model fields on the lower rows to be model-obs differences.

Thanks for your advice. We modified Fig. 12b. Now it shows differences between the RMSEs of SIC forecast results of Exp_SIC and Exp_SIC&SIT. The original Fig. 13 has also been modified, and the new Fig. 13 shows the difference between original Fig. 13a and Fig. 13b. Other figures with similar problems have also been modified.

**Typos and technical comments**

I attach and annotated version of the original pdf with technical comments.

I do not highlight all instances where the English language needs to be improved only the cases where the language is unclear to the point that the scientific understanding is inhibited.

Thank you very much for your detailed comments. We carefully revised all the contents. Your comments are more profound for our understanding of SIT, which is of great help to us.

Dawson. G., Landy, J., Tsamados, M., Komarov, A. S., Howell, S., Heorton, H., Krumpen, T.: A 10-year record of Arctic summer sea ice freeboard from CryoSat-2, Remote Sens. Environ., 268, 112744, https://doi.org/10.1016/j.rse.2021.112744, 2022.

Liang, X., Losch, M., Nerger, L., Mu, L., Yang, Q., & Liu, C.: Using sea surface temperature observations to constrain upper ocean properties in an Arctic sea ice-ocean data assimilation system, J. Geophys. Res.-Oceans, 124, 4727-4743, https://doi.org/10.1029/2019JC015073, 2019.

Mu, L., Losch, M., Yang, Q., Ricker, R., Losa, S. N., & Nerger, L.: Arctic-Wide sea ice thickness estimates from combining satellite remote sensing data and a dynamic ice-ocean model with data assimilation during the CryoSat-2 period, J. Geophys. Res.-Oceans, 123, 7763-7780, https://doi.org/10.1029/2018JC014316, 2018.

Yang, Q., Losa, S., Losch, M., Liu, J., Zhang, Z., Nerger, L., Yang, H.: Assimilating summer sea-ice concentration into a coupled ice-ocean model using a LSEIK filter. Ann. Glaciol., 56, 38-44, https://doi.org/10.3189/2015AoG69A740, 2015a.

Yang, Q., Losa, S., Losch, M., Jung, T., Nerger, L.: The role of atmospheric uncertainty in Arctic summer sea ice data assimilation and prediction. Q. J. R. Meterorol. Soc., 141: 2314-2323, https://doi.org/10.1002/qj.2523, 2015b.